# Unfolding the multiscale structure of networks with dynamical Ollivier-Ricci curvature

Adam Gosztolai [1✉] & Alexis Arnaudon[2,3]

Describing networks geometrically through low-dimensional latent metric spaces has helped design efficient learning algorithms, unveil network symmetries and study dynamical network processes. However, latent space embeddings are limited to specific classes of networks because incompatible metric spaces generally result in information loss. Here, we study arbitrary networks geometrically by defining a dynamic edge curvature measuring the similarity between pairs of dynamical network processes seeded at nearby nodes. We show that the evolution of the curvature distribution exhibits gaps at characteristic timescales indicating bottleneck-edges that limit information spreading. Importantly, curvature gaps are robust to large fluctuations in node degrees, encoding communities until the phase transition of detectability, where spectral and node-clustering methods fail. Using this insight, we derive geometric modularity to find multiscale communities based on deviations from constant network curvature in generative and real-world networks, significantly outperforming most previous methods. Our work suggests using network geometry for studying and controlling the structure of and information spreading on networks.

[1] Neuroengineering Laboratory, Brain Mind Institute & Interfaculty Institute of Bioengineering, EPFL, Lausanne, Switzerland. [2] Department of Mathematics, Imperial College London, London, UK. [3] Blue Brain Project, École polytechnique fédérale de Lausanne (EPFL), Geneva, Switzerland. ✉email: adam.gosztolai@epfl.ch

Real-world networks are rarely embedded in physical or Euclidean spaces, which complicates their analysis. Therefore, previous works have typically assumed that the network's nodes lie in a latent metric space[1]. A well-chosen metric space can provide a 'geometric backbone' to allow the correct representation of node similarities, and to study symmetries and dynamical processes on networks at a fundamental level. For example, assuming an underlying manifold structure[2] permits the efficient functioning of clustering algorithms based on Euclidean geometric features such as $k$-means or expectation maximisation[3]. Similarly, the hyperbolic space of constant negative curvature provides a natural parametrisation of complex networks to unveil their self-similar clusters across scales[4,5]. Besides, networks can also be embedded based on a suitable pseudo-distance metric between dynamical network processes, which has helped reveal their functional organisation[6,7]. However, in general, there is no guarantee that a network is compatible with a given metric space without suffering significant distortion[8]. Yet, a network may have several, not necessarily self-similar, geometric representations owing to multiscale structure, arising, for example, from clusters at multiple resolutions[9]. Thus, there is a need for a geometric notion that does not rely on predefined embedding spaces, yet allows unfolding the multiscale structure of a general class of networks.

A promising alternative to embeddings is to define geometry based on a notion of curvature, such as the Ollivier–Ricci (OR) curvature[10], which intuitively speaking measures the deviation of the graph from being locally grid-like analogously to being 'flat' in continuous spaces. 'Flatness' of a network can be understood in terms of its local connectivity: the distance of a pair of nodes is the same as the average distance of their neighbourhoods. Thus positive (or negative) OR curvature of an edge indicates that it resides in a region of the graph that is more (or less) connected than a grid. In addition to the intuitive definition, the OR curvature does not impose geometry through embedding but induces an effective network geometry with a precise interpretation in limiting cases. In fact, it is the only one among several discrete curvature notions[11,12] known to converge rigorously to the Ricci curvature of a Riemannian manifold[13]. The OR curvature has also been linked to graph-theoretical objects by deriving formal bounds on the local clustering coefficient and the spectrum of the graph Laplacian[14,15]. Moreover, the OR curvature of an edge is intrinsically linked to network-level robustness to edge removal, which has led to advances in applications such as studying the fragility of economic networks[16] or characterising the human brain structural connectivity[17].

However, despite recent clustering heuristics based on the OR curvature[18,19], several of its properties have hindered its widespread adoption to study network clusters. Firstly, since the OR curvature depends on structural neighbourhoods, related clustering methods (including the Ricci flow method[19]) lack a resolution parameter to tune the geometry to unveil multiscale structure in real-world networks. Multiscale clustering has been the subject of intense research and several methods and heuristics have been proposed, along with a parallel list of goodness measures for community structures. These include, without claim of exhaustivity, methods based on statistical mechanical models[20,21], normalised cut[22], nonnegative matrix factorisation[23], modularity[24–26] and extensions thereof using random walks and diffusion processes[9,27] as well as methods based on graph signal processing[28]. The second shortcoming of the classical OR curvature of an edge is that it is a local quantity, which depends on the degrees of its endpoints[14]. Thus, it likely provides a sub-optimal geometric representation of sparse networks—including many real-world networks where each node connects only to a few others—in which node degrees vary widely. In fact, the

classical OR curvature is related to the spectral gap of the graph Laplacian[15], the central object of spectral clustering methods[29], which no longer indicates clusters in sparse graphs[30], similarly to other nodes clustering methods[24]. This lack of robustness of the OR curvature for sparse networks also precludes its use for studying the limit of information spreading in graphs[31], which is linked to a phase transition occurring as the community structure gets weaker and becomes abruptly undetectable[31–33].

In other words, there is a need for a geometric notion that does not rely on embeddings, is capable of generating a family of geometric representations to encode multiscale clusters as increasingly coarser features. Moreover, these features should robustly signal network clusters at different scales until the fundamental limit of their detection. Such a notion would hold the premise to describe multiscale structures of graphs without the need for statistical null models and to open new avenues to study and control information spreading phenomena using network geometry.

## Results

**Dynamical OR curvature from graph diffusion**. We address this need by combining two distinct frameworks—network-driven dynamical processes and geometry with OR curvature. The spreading of network-driven dynamical processes is shaped by the heterogeneity of the network connectivity. In turn, one may infer the network structure by observing properties of their evolution. We focus on Markov diffusion processes[9,34–36], a class of linear dynamical systems which is rich enough to capture several properties of nonlinear processes on networks[37,38]. Let us consider a connected network of $n$ nodes and $m$ edges weighted by pairwise distances $w_{ij}$. We construct a continuous time diffusion on the network by the standard procedure[29] of defining the normalised graph Laplacian matrix $\mathbf{L} := \mathbf{K}^{-1}(\mathbf{K} - \mathbf{A})$, where $\mathbf{K}$ is the diagonal matrix of node degrees with $K_{ii} = \sum_j A_{ij}$ and $\mathbf{A}$ is the weighted adjacency matrix encoding similarities between nodes. To obtain non-negative similarities from node-to-node distances, one may simply take $A_{ij} = \max_{uv} w_{uv} - w_{ij}$ or $A_{ij} = e^{-w_{ij}}$, with the latter more strongly penalising distant points. Then, the probability measure of the diffusion started from the unit mass $\delta_i$ on node $i$ (Fig. 1a, b) evolves according to

$$\mathbf{p}_i(\tau) = \delta_i e^{-\tau \mathbf{L}} . \tag{1}$$

In analogy to the Ricci curvature on a manifold, the classical OR curvature[10,39] measures the distance of one-step neighbourhoods of a pair of nodes $i, j$ relative the geodesic (shortest path) distance of $i, j$ (see Supplementary Note 1 for background). Here, instead of structural neighbourhoods we consider distributions generated by diffusion processes across scales $\tau$. Specifically, we start a diffusion process at each node $i = 1, \ldots, n$ to obtain a set of measures $\mathbf{p}_i(\tau)$. We then define the *dynamic* OR curvature of an edge as the distance of the pair of measures started at its endpoints relative to the weight of the edge

$$\kappa_{ij}(\tau) := 1 - \frac{\mathcal{W}_1(\mathbf{p}_i(\tau),\ \mathbf{p}_j(\tau))}{w_{ij}} , \tag{2}$$

whenever $ij$ is an edge and 0 otherwise. Intuitively, Eq. (2) reflects the overlap of diffusions over time when started $w_{ij}$ distance apart, measured by $\mathcal{W}_1$, the optimal transport distance[40]. The latter is obtained as a solution to a minimisation problem (Eq. (11) in the "Methods" section) yielding the least cost of transporting the measure $\mathbf{p}_i(\tau)$ to $\mathbf{p}_j(\tau)$ by the optimal transport plan $\zeta(\tau)$. The entries of the optimal transport matrix are shown on Fig. 1c, d representing the quantity of mass moved between

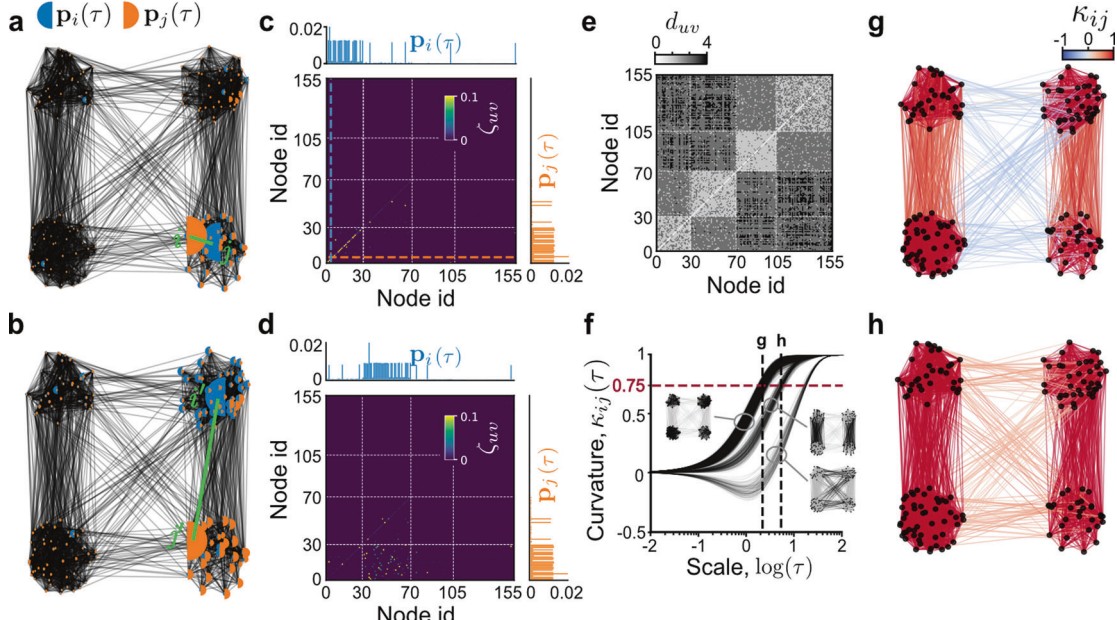

**Fig. 1 Dynamical Ollivier–Ricci curvature capturing the spreading of diffusion processes. a** Snapshot at time $\log \tau = 0.15$ of a pair of diffusion measures $\mathbf{p}_i(\tau)$ and $\mathbf{p}_j(\tau)$ started at nodes $i, j$ of a stochastic block model network (with four clusters of sizes 30, 40, 35, 50 and respective edge probabilities 0.7, 0.8, 0.9, 0.6 within clusters, and 0.1, 0.02 between clusters). When $i$ and $j$ are in the same cluster, the measures overlap significantly. The size of half-circles is proportional to the amount of mass on the respective nodes. **b** For $i'$, $j'$ in different clusters the measures remain largely disjoint. **c** Optimal transport plan $\boldsymbol{\zeta}$ $(\tau)$ superimposed with $\mathbf{p}_i(\tau)$, $\mathbf{p}_j(\tau)$. When $i, j$ (coloured dashed lines) lie in the same cluster only diagonal elements $\zeta_{uu}$ are positive, meaning only geodesics within a cluster transport significant mass. The white dashed lines correspond to the four clusters. **d** When the diffusions started at nodes $i'$ and $j'$ in different clusters only entries $\zeta_{uv}$ with $u$ and $v$ in the corresponding clusters have significant nonzero weight. **e** Geodesic distance matrix $d_{uv}$ showing the block structure of the network. **f** The evolution of the edge curvatures $\kappa(\tau)$ (Eq. (2)) against time. There are three distinct curvature bundles and in the insets we manually highlighted the edges to reveal their position in the graph. The shade of grey within the bundles reflect the density of edges. Here $\kappa_{ij}(\tau) \simeq 0.75$ marked by the red dashed line indicates scales when local mixing occurs between the corresponding diffusion pairs. The dashed vertical lines show two such scales ($\log \tau = 0.15, 0.43$). **g**, **h** Graph edges coloured by the curvature reveals clusters at the two scales.

each pair of nodes $u$ and $v$ along their connecting geodesic of length $d_{uv}$ (Fig. 1e).

By contrast to Eq. (2), previous works have typically defined the classical OR curvature based on one-step transition probabilities of lazy random walks $\mathbf{p}_i(\tau) \simeq \mathbf{p}_i = \alpha \mathbf{I} + (1 - \alpha)\delta_i \mathbf{K}^{-1}\mathbf{A}$. In this definition, scale is introduced via a laziness (or idleness) parameter $\alpha$, varying the importance of local neighbourhoods relative to $w_{ij}$, which in effect introduces self-loops in the graph. Our definition (Eq. (2)) replaces one-step neighbourhoods by probability measures supported on the whole graph, with the timescale $\tau$ of the diffusion process playing the role of the scale parameter. As expected, Eq. (2) recovers the classical OR curvature[10] as a first-order approximation. In addition, the dynamical OR curvature inherits the geometric intuition of the classical definition. Analogously to the Ricci curvature on planes, spheres and hyperboloids, $\kappa_{ij}(\tau)$ is zero on grids but positive and negative on cliques-like and tree-like networks, respectively, for all finite scales $\tau$ (Supplementary Fig. 1b, c). In the following, we are interested in studying the curvature distribution across edges when the network structure deviates from these canonical topologies.

**Edge curvature gap differences in rate of information spreading.** Most real-world networks exhibit organisation on several scales. As an illustration Fig. 1a, b shows an unweighted stochastic block model (SBM) network[41] of four clusters and two nontrivial scales. To construct these scales, we drew edges independently between clusters with different probabilities (0.1 or 0.02). We varied cluster sizes and within-cluster edge probabilities but ensured that the latter remained sufficiently high to easily

visualise clusters (see Fig. 1 for parameters). We show that this multiscale structure can be revealed by scanning through a finite range of scales $\tau$ and studying snapshots of curvature distribution across edges.

The characteristic scales of a network are related to the overlap between pairs of diffusion measures $\mathbf{p}_i(\tau)$, $\mathbf{p}_j(\tau)$. This overlap depends on the starting points $i, j$ and on network clusters which can confine diffusions on well-connected regions for long times before reaching the stationary state $\boldsymbol{\pi}$[9,34,35,42], given by $\pi_i = \mathbf{K}_{ii}/\sum_i \mathbf{K}_{ii}$. This transient phenomenon is reflected by the structure of the optimal transport matrix $\boldsymbol{\zeta}(\tau)$. If $i, j$ lie within the same cluster, the measures quickly overlap (Fig. 1a) and only diagonal entries of $\boldsymbol{\zeta}(\tau)$ are positive (Fig. 1c), weighing only short, within-cluster geodesics. By contrast, started at different clusters, the measures remain almost disjoint (Fig. 1b) and $\boldsymbol{\zeta}(\tau)$ is forced to select longer geodesics (Fig. 1d, e), reflected by the large entries in the off-diagonal block.

The evolution of the edge curvature $\kappa_{ij}(\tau)$ (Fig. 1f) aggregates the information in $\boldsymbol{\zeta}(\tau)$ into a single number that is related to the rate of mass exchange between clusters at a given scale. We see in Fig. 1f that, initially, when all nodes support disjoint point masses and the diffusions have not yet mixed, $\lim_{\tau \to 0} \kappa_{ij}(\tau) \to 1 - \mathcal{W}_1(\delta_i, \delta_j)/d_{ij} = 0$. At the other extreme, as the diffusions reach stationary state, $\lim_{\tau \to \infty} \kappa_{ij}(\tau) \to 1 - \mathcal{W}_1(\boldsymbol{\pi}, \boldsymbol{\pi})/d_{ij} = 1$. At intermediate scales, the curvature can take values between 1 and some finite negative number depending on the graph[15]. We find that, as the curvature of an edge evolves, the scale at which it approaches unity indicates how easy it is to propagate information between clusters. More precisely, in the "Methods" section, we prove that this scale gives an upper bound on

the mixing time $\tau_{ij}^{\text{mix}}$ of the diffusion pair, namely,

$$
\begin{aligned}
\tau_{ij}^{\text{mix}} &:= \frac{1}{2}\sum_{uv}|\zeta_{uv}(\tau) - \zeta_{uv}(\infty)| \\
&\leq \min\{\tau : \kappa_{ij}(\tau) \geq 0.75\},
\end{aligned}
\tag{3}
$$

where $\boldsymbol{\zeta}(\tau)$ is the optimal transport plan with marginals $\mathbf{p}_i(\tau)$ and $\mathbf{p}_j(\tau)$. Note that $\kappa_{ij}(\tau) \geq 0.75$ does not imply that the corresponding diffusion processes have approached stationary state independently, but only that they exchange negligible mass at that or larger scales.

Importantly, a gap in the distribution of curvatures appears when the curvature exceeds 0.75 for some edges while being <0.75 for others indicating a network bottleneck that limits mass flow. To illustrate this, Fig. 1f shows three bundles of edges, the edges within the bundle with most positive curvature are found within clusters, while edges within the other two bundles lie between clusters. Figure 1g, h show the two scales on Fig. 1f (log $\tau$ = 0.15, 0.43) where the curvature has exceeded 0.75 for a given bundle, indicating the diffusions are well mixed across the corresponding edges, but not across other edges whose curvature is <0.75. The latter mark bottleneck edges which lie between the expected partitions with four and two clusters, respectively. This simple example shows the importance of the scale parameter $\tau$ in our curvature definition to capture network scales. Let us emphasise that the characteristic scales revealing network clusters in our framework are indicated by curvature gaps, i.e., differences in the relative magnitude of curvatures. This is unlike some previous works[18,19], where clusters were identified based on finding negatively curved edges between clusters. Before applying this idea to real networks, we take a closer look at the curvature gap in the theoretical context of the stochastic block model.

**Curvature gap is a robust indicator of clusters in stochastic block models.** Since in our example any pair of diffusions are supported by one (Fig. 1a) or two (Fig. 1b) clusters, we focus on the subgraph $G$ induced by two clusters (Fig. 2a). Let us simplify one more step and assume that $G$ is a realisation of $\mathcal{G} = \text{SSBM}(n/2, p_{\text{in}}, p_{\text{out}})$, the symmetric SBM composed of two planted partitions of equal size. Edges are generated independently with probability $p_{\text{in}}$ within-clusters and probability $p_{\text{out}}$ between-clusters. This symmetry assumption is not necessary in general, as illustrated by our other examples, but it allows us to make links to known theoretical results. We will denote the ground truth as $C_i^* \in \{1, -1\}$ for each node $i$ and define $\bar{k} = n(p_{\text{in}} + p_{\text{out}})/2$ as the average degree.

Classical spectral clustering methods[29] perform well for dense graphs (Fig. 2a), where $\bar{k}$ is an increasing function of $n$. This suppresses fluctuations for large $n$ causing a spectral gap to appear when the eigenvalue $\lambda_c$ of the Laplacian matrix $\mathbf{L}$ of $G$ separates from bulk eigenvalues arising from randomness[29] (Fig. 2c). In this dense regime, $\lambda_c$ is well approximated by $\langle\lambda_c\rangle_{\mathcal{G}} = 2p_{\text{out}}/(p_{\text{in}} + p_{\text{out}})$, the second eigenvalue of the ensemble averaged Laplacian $\langle\mathbf{L}\rangle_{\mathcal{G}}$ (see Supplementary Note 2). Since $\lambda_c$ can be identified due to the spectral gap, clustering involves simply labelling nodes by the sign of the entries of the corresponding eigenvector $\phi_c(u) = 1/\sqrt{n}$ when $C_u^* = 1$ and $-1/\sqrt{n}$ when $C_u^* = -1$. However, for sparse graphs (Fig. 2b), where $\bar{k}$ is constant (independent of $n$), the spectral gap ceases to exist[43] (Fig. 2d). Thus, spectral algorithms relying on identifying $\lambda_c$ perform no better than chance. To perform clustering in this regime, one needs to go beyond spectral clustering using, for example, the belief propagation method in statistical physics or the related non-backtracking operator whose spectrum is better behaved[31,33].

To see how robustly the dynamical OR curvature indicates the presence of clusters in the symmetric SBM, we define the curvature gap as the difference between the mean curvatures of within- and between-edges at a given scale

$$
\Delta\kappa(\tau) := \frac{1}{\sigma}\left|\langle\kappa_{ij}(\tau)\rangle_{C_i^* = C_j^*} - \langle\kappa_{ij}(\tau)\rangle_{C_i^* \neq C_j^*}\right|.
\tag{4}
$$

Here the averages are over within and between-edges, normalised by $\sigma = \sqrt{\frac{1}{2}\left(\sigma_{\text{within}}^2 + \sigma_{\text{between}}^2\right)}$ in terms of the standard deviations of both sets of curvatures. This measure is adapted from the sensitivity index in signal detection theory, known to be, asymptotically, the most powerful statistical test for discriminating two distributions[44]. Large curvature gap $\Delta\kappa(\tau)$ indicates that the within and between edges have curvatures different enough for the clusters to be recovered (Fig. 2e, f). Correspondingly, in the limits $\tau \to 0, \infty$ where the curvatures are uniform across the graph $\Delta\kappa(\tau)$ vanishes and, likewise, in the absence of structure ($p_{\text{in}} \approx p_{\text{out}}$ in the Erdős–Rényi (ER) limit) we have $\Delta\kappa(\tau) = 0$ for all $\tau$ (Fig. 2g). At intermediate scales, we find that the scale of maximal curvature gap occurs at $\tau_\kappa$ at which point the curvatures of within-edges is $\kappa_{ij}(\tau_\kappa) \approx 0.75$. In agreement with Eq. (3), this indicates well-mixed diffusions across these edges relative to low-curvature bottleneck edges between clusters, which indicate incomplete mixing. We also find that $\tau_\kappa \approx \lambda_c^{-1}$ (Fig. 2e, f). These results show that positive curvature gap is associated with the presence of clusters.

What is the minimum curvature gap needed to detect clusters? Previous works on the limits of cluster detection has shown that if the clusters are too weak (high $r := p_{\text{out}}/p_{\text{in}}$) or the graph too sparse (low $\bar{k}$), no clustering algorithm performs better than chance, or distinguish $G$ from an Erdős–Rényi graph ($r = 1$). This is known as the limit of weak-recovery or detection and is characterised by the Kesten–Stigum (KS) threshold $r = r_{\text{KS}} = (\bar{k} - \sqrt{\bar{k}})/(\bar{k} + \sqrt{\bar{k}})$[31,32,45].

To study this limit, we sampled 20 networks from $\mathcal{G}$ for a range of $\bar{k}$ and $r$. For each sample, we computed the maximal curvature gap $\Delta\kappa^* := \max_\tau \Delta\kappa(\tau)$ and formed the ensemble average quantity $\langle\Delta\kappa^*\rangle_{\mathcal{G}}$. As $r$ increases for a given $\bar{k}$ we observe that $\langle\Delta\kappa^*\rangle_{\mathcal{G}}$ decreases exponentially until a certain noise level (Fig. 2h). The critical edge density ratio $r_{\bar{k}}^*$ can be estimated as the smallest $r$ where $\langle\Delta\kappa^*\rangle_{\mathcal{G}}$ dropped below a threshold background noise level, estimated here at 0.035 (black horizontal line). This choice of threshold is not absolute, as it is affected by the finite-size effect of the SBM graphs. An analytical derivation of this threshold is out of scope of this work, but our numerical experiment clearly shows that the curvature gap detects a signal from the planted partitions up to the KS limit (Fig. 2i).

**Geometric cluster detection in the sparse regime.** Given that the curvature gap (Eq. (4)) indicates the presence of clusters until the fundamental KS limit, we asked if this information could be used to recover the ground truth partition. The definition of curvature gap (Eq. (4)) suggests looking for equilibrium configurations of the unit-temperature Boltzmann distribution over the cluster assignments $C$,

$$
\mathbb{P}(C|\boldsymbol{\kappa}) \propto e^{\sum_{ij}\kappa_{ij}(\tau)\delta(C_i, C_j)},
\tag{5}
$$

where $\boldsymbol{\kappa}$ is a matrix with entries $\kappa_{ij}$, the sum is over all edges $ij$ and $\delta(C_i, C_j) = 1$ if $C_i = C_j$ and 0 otherwise. The distribution involves only within-edges because finding those is equivalent to finding between-edges, up to a normalisation factor.

The distribution $\mathbb{P}(C|\boldsymbol{\kappa})$ is important because all of its equilibrium states are equivalent and correlate with the ground truth partition of the symmetric SBM $\mathcal{G}$. To see this, we connect $\mathbb{P}(C|\boldsymbol{\kappa})$ to the posterior distribution $\mathbb{P}(C|G)$ of the cluster

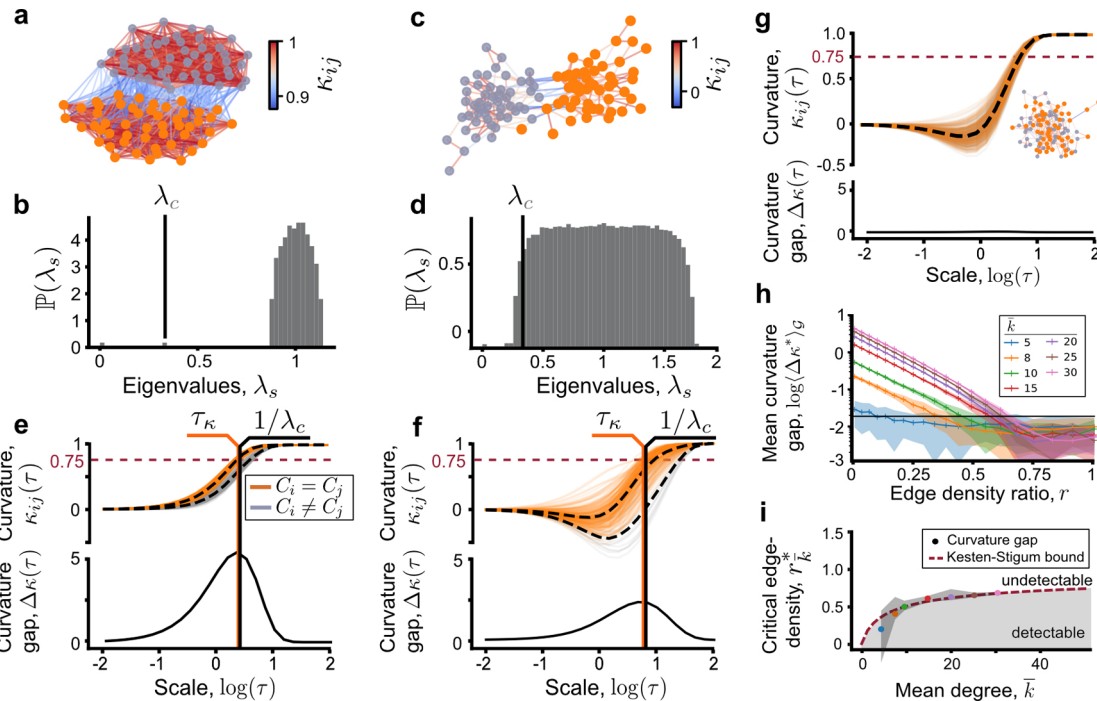

**Fig. 2 Edge curvature gap indicates the presence of clusters where spectral clustering fails.** Two-partition symmetric SBM graph in the **a** dense regime ($p_{in} = 0.5$, $p_{out} = 0.1$) and **b** sparse regime ($p_{in} = 8/n$, $p_{out} = 0.5/n$). Edges are coloured by the curvature ($n = 100$, $\log \tau = 0.83$). **c, d** The histogram of eigenvalues obtained from five SBM realisations in dense and sparse regime, respectively. In the dense regime, the eigenvalue $\lambda_c$ corresponding to the community structure is well separated from the bulk eigenvalues, but overlaps in the sparse regime. **e, f** The evolution of edge curvatures driven against diffusion time. A gap between the curvatures of within-edges (orange) and between-edges (grey) is associated with the presence of clusters. When $\kappa_{ij} > 0.75$ (horizontal dashed line) the diffusions are well mixed across the respective edges. The curvature gap is maximal at $\tau_\kappa \approx \lambda_c^{-1}$ (orange and black vertical lines). **g** There is no curvature gap in the limiting ER graph (inset, $p_{in} = p_{out} = (8 + 0.5)/(2n)$). **h** Maximal curvature gap averaged over 20 SBM realisations for each fixed $\bar{k}$ with $10^4$ nodes, against edge density ratio. The horizontal line marks the estimated background noise level. The intersection of this line with the mean curvature gap defines $r_{\bar{k}}^*$, the largest possible edge density ratio to detect clusters. Shaded error bars indicate one standard deviation from the mean. **i** Phase diagram of critical edge density ratio against average degree. The numerically obtained critical edge density ratios computed from the curvature gap are superimposed with the theoretical Kesten–Stigum detection limit (dashed line) and show excellent agreement. Shaded error bars indicate one standard deviation from the mean. Grey shaded area denotes the regime where detection is possible.

assignments given the graph drawn from $\mathcal{G}$. In the sparse regime, the likelihood of observing $G$ with a given cluster assignment $C$ is

$$\mathbb{P}(G|C) \propto \prod_{ij} \left( \frac{p_{in}}{p_{out}} \right)^{\delta(C_i, C_j)} \propto \mathbb{P}(C|G) \qquad (6)$$

(see Eq. (15) in the "Methods" section). The second part of Eq. (6) results from Bayes' theorem using a uniform prior on $C$, since a priori all configurations are equally likely. It has been previously shown[31] that $P(C|G)$ is equivalent to the Boltzmann distribution of an Ising model with constant interaction strength

$$\mathbb{P}(C|G) \propto e^{\beta \sum_{ij} \delta(C_i, C_j)} \qquad (7)$$

with inverse temperature $\beta = \log(p_{in}/p_{out}) \approx p_{in} - p_{out}$. Note that the equilibrium state $(1, 1, \ldots, 1)$ is trivial assigning all nodes to one cluster. However, asymptotically ($n \rightarrow \infty$) the probability of this state vanishes and the Boltzmann distribution is uniform over all other configurations with group sizes $n/2$ and $p_{out} n/2$ between-edges[31]. The fact that one of these states is the ground truth partition, and all equilibrium states of Eq. (7) are equivalent up to a permutation of nodes within clusters means they are indistinguishable from the ground truth partition.

Due to the equivalence between Eqs. (6) and (7), to prove the equivalence between Eqs. (5) and (6) we show that Eq. (5) can also be reduced to Eq. (7). The main insight is that the dynamical OR curvature (Eq. (2)) is constructed using pairs of diffusions, as opposed to single diffusions used in previous studies[9,20,35]. Thus,

eigenmodes arising from random fluctuations, which would otherwise confound methods relying on the spectrum of the Laplacian, are reflected equally in the spectrum of both diffusions and cancel out upon taking differences over all adjacent node pairs. This allows recovering the community eigenvector $\phi_c$ even in the sparse regime where the spectral gap vanishes and $\lambda_c$ is no longer identifiable from the spectrum (Fig. 2d). To see this, we consider the difference between a pairs of diffusions and use the spectral expansion to write $\sum_{ij}(p_i^u(\tau) - p_j^u(\tau)) = \sum_s e^{-\lambda_s \tau} \phi_s(u) \Delta \phi_s$ where

$$\Delta \phi_s := \sum_{ij} \left( \phi_s(i) - \phi_s(j) \right) . \qquad (8)$$

We find that, instead of looking at the eigenvalue distribution (Fig. 2d), the community eigenvector $\phi_c$ can be recovered by the relative amplitude of $\Delta \phi_s$. Indeed, on a single SBM realisation, $\Delta \phi_s$ is large for only a few eigenvectors $\phi_s$ and diminishes for others Fig. 3a. Importantly, those and only those eigenvectors with large $\Delta \phi_s$ correlate strongly with the ground truth (Fig. 3a inset). As seen in Fig. 3b, the best eigenvector is not $\phi_2$, i.e., the one whose eigenvalue is second in the spectrum and is used by spectral clustering methods, but the one whose eigenvalue is inside the bulk in Fig. 2d and thus cannot be identified by looking at the spectrum alone. The correlation with the ground truth for $\phi_c$ with the highest $\Delta \phi_s$ averaged over 50 SBM realisations remains close to the highest achievable among all eigenvectors as the KS bound is approached. Meanwhile, $\phi_2$, the eigenvector used by spectral

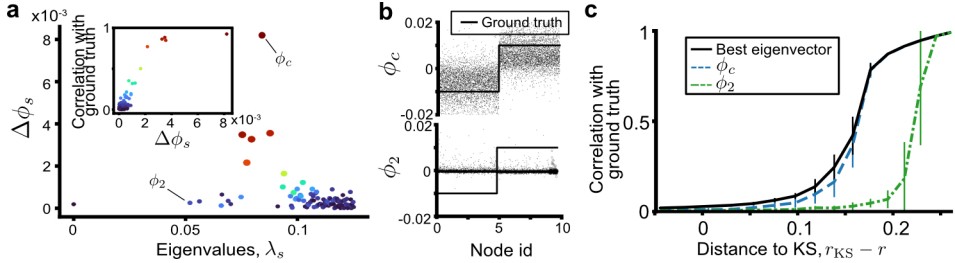

**Fig. 3 Detecting communities using pairs of diffusions near the weak recovery limit. a** Difference in eigenvectors $\Delta\phi_s$ (Eq. (8)) between diffusion processes started at adjacent nodes for a single sparse SBM network ($p_{in} = 3/n$, $p_{out} = 0.5/n$, $n = 10^4$). Each dot marks ($\lambda_s$, $\Delta\phi_s$) for the 50 smallest eigenvectors, coloured by the correlation of the corresponding eigenvector with the ground truth, shown in the inset. **b** The eigenvector $\phi_c$ with the highest $\Delta\phi_s$ encodes the cluster structure (solid line), whereas the second eigenvector $\phi_2$, used by spectral clustering methods, are driven by high random fluctuations. **c** Correlation of eigenvectors with ground truth against distance to KS limit ($n = 10^5$, $\bar{k} = 3$). The eigenvector identified by the highest $\Delta\phi_s$ approaches the correlation with the ground truth of the best eigenvector in the spectrum. All eigenvectors become uncorrelated with the ground truth at the KS limit. Error bars indicate one standard deviation from the mean.

clustering methods is suboptimal (Fig. 3c). We also found that, close to the KS bound, often a few other eigenvectors with similarly high $\Delta\phi_s$ appear, suggesting an improved clustering method combining several top eigenvectors, but this is out of scope here.

To express the curvature in the exponent of Eq. (5) we use the dual formulation of the optimal transport distance (Eq. (12) in "Methods" section). The fact that $\Delta\phi_c$ dominates the contribution from other eigenvectors, allows us to approximate $\sum_{ij}(p_i^u(\tau) - p_j^u(\tau)) = e^{-\lambda_c\tau}\phi_c\Delta\phi_c + \epsilon_\phi \propto e^{-\lambda_c\tau}\phi_c + \epsilon_\phi$, where $\epsilon_\phi$ is an asymptotically small term. We use this expression, together with the duality formula (Eq. (12)) to express Eq. (5). Finally, in the sparse regime, we may make a tree-like approximation of the neighbourhoods of $i$ and $j$ to find that Eq. (5) reduces to

$$\mathbb{P}(C|\kappa) \propto e^{|p_{in}-p_{out}|\sum_{ij}\delta(C_i,C_j)}. \tag{9}$$

We refer the reader to the "Methods" section for details. Eq. (9) is the same as Eq. (7) when the communities are assortative ($p_{in} > p_{out}$). We then conclude that the curvatures encode the communities of the symmetric SBM and allow it to be recovered until close to the Kesten–Stigum bound.

In the next section, we present a clustering algorithm based on this insight that can find multiscale clusters in real-world networks.

**Geometric modularity for the multiscale clustering of networks.** To exploit the property of the dynamical OR curvature to give multiple geometric representations, we develop a multiscale graph clustering algorithm for real-world networks. Using Eq. (5), we introduce the geometric modularity function

$$Q_\kappa(C,\tau) = \frac{1}{2m_\kappa}\sum_{ij}(\kappa_{ij}(\tau) - \kappa_0)\delta(C_i, C_j), \tag{10}$$

where $2m_\kappa = \sum_{ij}|\kappa_{ij}|$ is a normalisation factor and $\kappa_0 = \max_{ij}\kappa_{ij}(\tau_{min})$ is a constant ensuring that all edges have small non-positive curvature at the smallest computed scale $\tau_{min}$. Hence optimising Eq. (10) at small times yields separate communities for each node whereas at large times, when $\kappa_{ij}(\tau) \to 1$ for all $ij$, all nodes are merged to a single community. At intermediate scales, the curvatures will have negative and positive values on different edges, making the detection of non-trivial clusters possible without a statistical null-model. This is in contrast to classical modularity[24], which minimises the expected number of edges between clusters, and requires a statistical null-model (typically the configuration model), which can hinder identifying functional communities based on dynamics[6].

To detect robust partitions at several scales, we compute the curvature distribution at scales $\tau$ spanning the entire dynamical range of the curvature and, at each $\tau$, we sample the cluster landscape $Q_\kappa(C,\tau)$ by optimising Eq. (10) using the Louvain algorithm[46,47] with $10^2$ random initialisations. At a given $\tau$, we take the cluster with the highest geometric modularity and deem it robust if it has a low variation of information $VI_\tau$ against 50 other randomly chosen clusters at this scale, as well as low variation of information $VI_{\tau\tau'}$ against the best cluster assignments at nearby scales $\tau'$. As an example, we show in Fig. 4a the result of this computation on our four-partition SBM graph with two hard-coded scales. We clearly see two large plateaus with low $VI_\tau$ and $VI_{\tau\tau'}$, corresponding to robust clusters, shown in Fig. 4b, c. At the smallest scales we find no robust communities shown by the sharp increase in the number of communities and the large $VI_\tau$. We compared geometric modularity to other clustering methods on the SBM and LFR generative benchmark graphs achieving near-state-of-the-art accuracy in both cases, close to the theoretical limit (see Supplementary Fig. 2 and Supplementary Note 3 for details). Notably, our method performs substantially better than the Ricci flow[19] method based on the classical OR curvature reinforcing our theoretical insight that combining diffusion processes and OR curvature allows surpassing the limitations of previous OR curvature-based methods.

Our algorithm involves several steps including computing the geodesic distance matrix, computing the diffusions (Eq. (1)) starting from all nodes, computing the curvatures (Eq. (2)) for all edges and running the Louvain algorithm. We discuss the complexity in detail in Supplementary Note 5. Briefly, the step with highest complexity is the computation of the edge curvatures which using exact algorithms runs in time $O(mn^{5/2})$, which denotes that the computation time grows at most as $Mn^{5/2}$ for a positive constant $M$ and for $m, n$ sufficiently large. This complexity arises since the computation involves solving a linear programme of complexity $O(n^{5/2})$ for each of the $m$ edges. On sparse networks this is on par with other random walk[9,20] and probabilistic[21,25,27] methods. However, for small times a complexity close to $O(mn)$ can be achieved by 'trimming' the probability measures, i.e., reducing their support size by ignoring the mass on nodes below a certain cutoff value (Supplementary Fig. 4b). For large times, a complexity reduces to $O(mn)$ by replacing the the optimal transport distance by the regularised Sinkhorn distance[48] (Supplementary Fig. 4c). This computational speedups together with the parallelised implementation of algorithm means that it scales well to moderately sized graphs (~$10^4$ nodes).

Due to the link between high edge curvature and well-mixed state (Eq. (3)), we expected that at robust scales the clusters will

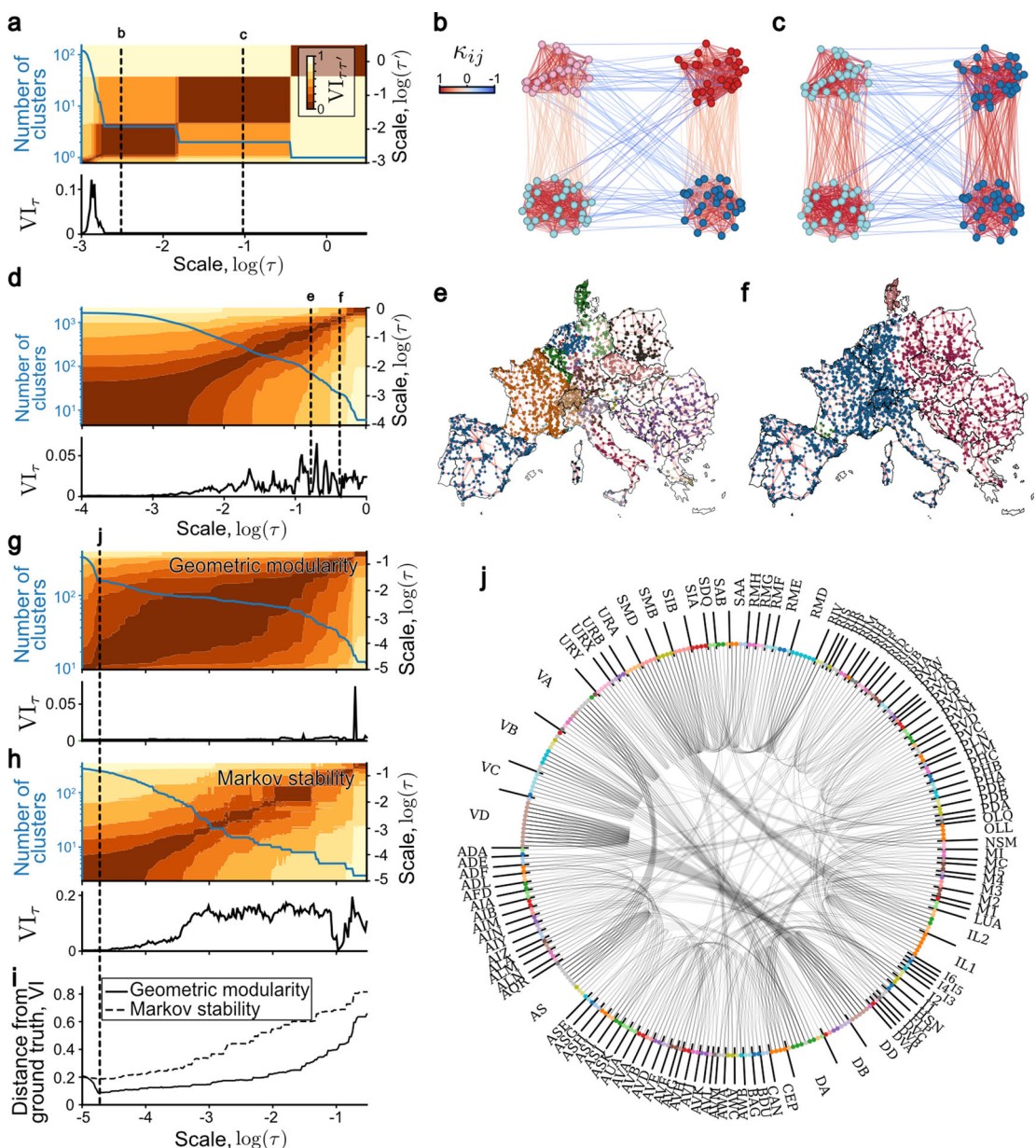

**Fig. 4 Clustering networks based on multiscale geometric modularity. a** Clustering statistics computed based on $10^2$ Louvain realisations for the multiscale stochastic block model graph. Vertical dashed lines show scales at which stable clusters are detected based on low variation of information at a given scale and persistent low variation of information between Louvain realisations across scales. The communities obtained are shown on **b** for scales $\log \tau = -2.3$ and **c** for $\log \tau = -1$. Edges are coloured by the curvatures $\kappa_{ij}(\tau)$ at the respective scales. **d** Clustering statistics for the European power grid. Two representative stable scales are shown **e** for $\log \tau = -0.95$ and **f** for $\log \tau = -0.5$. Node colours show community membership. **g** Clustering statistics for and the network of *C. elegans* single-neuron homeobox gene expressions show a plateau of stable scales with very similar partitions. **h** Clustering statistics obtained with Markov stability shows stable scales only at small times with single-node communities, indicating overfitting, and many non-robust partitions at larger scales with high variation of information. **i** Distance from ground truth based on structural neuronal types or the predicted clusters. Geometric modularity obtained significantly better performance than Markov stability. **j** Clustering of the *C. elegans* homeobox gene expression data obtained from geometric modularity optimisation superimposed with the ground truth.

correspond to those regions which have a high amount of redundant information, and thus can be disconnected without affecting the dynamics within them. To see this, we applied this clustering algorithm to the European power grid graph in Fig. 4d–f, an unweighted network of major electrical lines, which has been previously analysed for robustness[49], multiscale communities[50] and centrality[36]. The multiscale community structure can be clearly seen with the many minima of the $VI_\tau$ function in Fig. 4d. We displayed two scales in Fig. 4e, f which unfold parts of the power grid which have been historically

independently developed. The smaller scale (at around $\log \tau = -0.95$) marks economical or historical unions and states (Skandinavia, Benelux, Czechoslovakia, Balkans, etc.). Likewise, the larger scale (at around $\log \tau = -0.5$) divides historical Eastern–Western Europe. Interestingly with the boundary in Germany runs along the iron curtain, which also demarcates the regions between major electricity companies.

Finally, we analysed a recent dataset of homeobox gene expression in single neurons of *C. elegans*[51]. The authors in ref. [51] found based on a multivariate linear regression that the

homeobox gene expression profile in a given anatomical neuron class can explain on average 74% of the expression level of the remaining genes in that neuron class. We therefore asked whether the homeobox gene expression profile has sufficient information to cluster neurons into their known anatomical classes.

The data contains a binary feature vector for each of the 301 neurons, indicating the presence of a protein expressed by any of the 105 homeobox genes in the given neuron. To convert this data into a graph with nodes being neurons, we first eliminated all homeobox genes co-expressed in none or more than 90% of the neurons to retain 67 homeobox genes. We then constructed an all-to-all graph adjacency matrix weighted by the Jaccard similarity index between expression profiles of neurons. To increase the number of edges with negative curvature, thus improve the detection at the smallest scales, we sparsified this network using a geometric sparsification method[52] with parameter $\gamma = 0.01$. This method retains at most a fraction $\gamma$ edges of the original graph as minimum spanning tree augmented by edges relevant for preserving local or global geometry of the graph.

The results of our clustering algorithm on this graph is shown in Fig. 4g and compared with the result of Markov stability[9] on Fig. 4h, a multiscale method based on persistence of diffusions. Geometric modularity obtains a large range of robust scales with highly similar clusters—as shown by the low $VI_\tau$ and $VI_{\tau\tau'}$. These scales correlate closely with the known ground truth of 117 anatomical neuron classes (Fig. 4i). In contrast, for Markov stability[9], the scales with low $VI_\tau$ overfit the graph finding too many clusters (Fig. 4h) which correlate less with the ground truth (Fig. 4i). Likewise, hierarchical clustering fails to identify the ground truth communities[51]. We also compared our result with that obtained from a broad range of clustering methods finding that other methods either overfit the neuron classes or found too few partitions (Supplementary Note 4 and Supplementary Fig. 3). Althouth the wavelet method of Tremblay and Borgnat returned a clustering near the ground truth, this scale was not identifiable based on their stability metric (Supplementary Fig. 3b). On Fig. 4j we superimpose the best clustering from geometric modularity against the ground-truth. We observe little differences, apart from VA and AS nodes as well as VD and DD often clustered together. Careful look reveals close biological relationship between these classes; all four classes correspond to motor neurons, with pairs expressing the same neurotransmitters—VA, AS expressing acetylcholine and VD, DD expressing gamma-aminobutyric acid (GABA). These novel results give direct quantitative support to the claim that homeobox gene expression patterns encode structural neuron types. We also observe other stable partitions at larger scale, but they did not correlate the ground-truth.

Overall, these results give a strong demonstration that our method is able to find stable clusters in sparse graphs, and provide meaningful insights into distinct types of real-world networks.

**Discussion**. Real-world networks often exhibit community structure on multiple scales, due to differences between the rates of information propagation in regions the network on various timescales. We introduced the concept of dynamical OR curvature which defines a scale-dependent geometry from the evolution of pairs of diffusion processes on the network. We showed that the edge curvature carries a precise meaning in this context bounding the rate of information flow across edges. Consequentially, gaps in the edge curvature distribution arising from differences between edge curvatures within and between regions indicate network bottlenecks. Systematically finding these gaps in the edge curvature distribution captures progressively coarser community features as the diffusion processes evolve. This result

does not rely on the dynamics being linear diffusions, making it suitable to study the interaction of arbitrary dynamical processes. We expect that, in the future, this approach can be used to tune the geometry of the graph to control the flux or interaction of network-driven dynamical processes, for example, leading to insights to metapopulation models[53] and synchronisation problem, for example, to better understand the coexistence of chimera states[54,55].

Unlike previous geometric approaches, which rely on embedding a network into a particular latent metric space[5–7,35], our approach constructs an effective object - the weighted and signed edge curvature matrix. Whilst not requiring specific assumptions used by latent space approaches it is worth noting that the dynamical OR curvature is constructed on the metric space formed by all the shortest paths of the graph. This property suggests links with the field of fractal geometry which studies scaling properties of graphs using the shortest path metric[56]. Thus, in graph families such as complex networks whose fractal geometry can be characterised[57] one can expect relationships between coarse-graining schemes based on box-covering techniques and aggregating clusters based on similar dynamical OR edge curvatures, which could be exploited for controlling the multifractal geometry of these networks.

Although diffusion processes constructed from the graph Laplacian have been explored for network clustering[9,27,35], our work differs in the use of diffusion pairs, as opposed to single diffusions, to construct the curvature. Diffusion pairs are implicitly coupled through the graph and pick up random variations independently, which can be exploited to average out non-informative fluctuations. On stochastic block models, this feature allows the curvature gap to robustly indicate clusters in the sparse regime down to the fundamental limit, where clustering methods relying on the spectral gap in the Laplacian fail[43]. We also found a new measure of eigenvalue quality, able to select the best eigenvector to be used in spectral methods. Interestingly, the edge curvatures are defined on the set of shortest paths which cannot contain the same edge twice, a subset of the set of non-backtracking walks. Our results are therefore consistent with previous works on the limits of cluster detection using statistical physics objects including the spectrum of non-backtracking operator[33] or related message passing approaches[31]. We expect this insight to provide a new avenue to study the fundamental limits of efficient clustering from a geometric perspective.

Finally, we introduced the notion of geometric modularity to build an easy-to-use multiscale clustering algorithm. Notably, our algorithm achieved near-state-of-the-art performance in sparse SBM graphs, better than methods relying on the spectral gap in the Laplacian matrix (e.g., spectral clustering[29] and edge-betweenness[25]) as well as those relying on the classical OR curvature[19]. This confirms that combining diffusions and OR geometry allows surpassing the limitations of these methods, which work well only on dense graphs. We also found robust and interpretable communities on multiple scales in real-world networks without the tendency of overfitting. Overall, we expect our insights connecting dynamical processes, geometry and network clustering to open new avenues to studying and controlling the structural and dynamical properties of networks.

## Methods

**Optimal transport distance**. To measure the distance between a pair of measures $\mathbf{p}_i(\tau)$ and $\mathbf{p}_j(\tau)$ we use the optimal transport distance[40] (also known as 1-Wasserstein or earth-mover distance), defined as

$$\mathcal{W}_1(\mathbf{p}_i(\tau), \ \mathbf{p}_j(\tau)) = \min_{\zeta} \sum_{uv} d_{uv} \zeta_{uv} \ ,$$
$$\text{subject to } \sum_v \zeta_{uv} = p_i^u(\tau) \ , \quad \sum_u \zeta_{uv} = p_j^v(\tau) \ . \tag{11}$$

The constraints in Eq. (11) ensure that the optimal transport plan $\zeta(\tau) \in \mathbb{R}^{n \times n}$ is a coupling of the measures $\mathbf{p}_i(\tau)$, $\mathbf{p}_j(\tau)$, i.e., $\zeta(\tau)$ is a joint distribution that admits $_{\mathbf{p}_i(\tau)}$ and $\mathbf{p}_j(\tau)$ as marginals.

An equivalent formulation of this distance can be constructed from the Kantorovich–Rubinstein duality[40], given by

$$\mathcal{W}_1(\mathbf{p}_i(\tau), \ \mathbf{p}_j(\tau)) = \sup_f \sum_u f(u)[p_i^u(\tau) - p_j^u(\tau)] \qquad (12)$$

where the supremum is taken over all 1-Lipschitz functions $f$ on the graph, that is,

$$|f(u) - f(v)| \le d_{uv} \qquad (13)$$

for any node pair $u, v$.

**Upper bound on the mixing time in terms of curvature**. Here we prove inequality (3), which gives an upper bound on the mixing time of the coupled diffusions with measures $p_i(\tau)$, $p_j(\tau)$ in terms the dynamical OR curvature. The $\epsilon$-mixing time is defined as the smallest $\tau$ where the law of the coupled process, the optimal transport plan $\zeta(\tau)$, is within an $\epsilon$ radius of the stationary distribution

$$\tau_{ij}(\epsilon) := \min\{\tau : \ ||\zeta(\tau) - \zeta(\infty)||_{\mathrm{TV}} \le \epsilon\}, \qquad (14)$$

where the notion of "close to stationarity" is quantified by the total variation distance $||\zeta(\tau) - \zeta(\infty)||_{\mathrm{TV}} := \frac{1}{2}\sum_{uv}|\zeta_{uv}(\tau) - \zeta_{uv}(\infty)|$. Since $p_i(\tau)$ and $p_j(\tau)$ are marginals of $\zeta(\tau)$ we have that

$$\tau_{ij}(\epsilon) = \min\{\tau : \ ||\mathbf{p}_i(\tau) - \boldsymbol{\pi}||_{\mathrm{TV}} + ||\mathbf{p}_j(\tau) - \boldsymbol{\pi}||_{\mathrm{TV}} \le \epsilon\}$$
$$= \min\{\tau : \ ||\mathbf{p}_i(\tau) - \mathbf{p}_j(\tau)||_{\mathrm{TV}} \le \epsilon\},$$

where we used the independence of the diffusion processes. From here, we may follow ref. [58] and use the Csiszár-Kullback-Pinsker inequality for the optimal transport distance

$$||\mathbf{p}_j(\tau) - \mathbf{p}_j(\infty)||_{\mathrm{TV}} \le (1/d_0)\mathcal{W}_1(\mathbf{p}_j(\tau), \ \mathbf{p}_j(\tau)) ,$$

where $d_0 = \min_{ij} d_{ij}$ is a global graph constant, which can therefore be absorbed into $\epsilon$. This gives an upper bound

$$\tau_{ij}(\epsilon') \le \min\{\tau : \ \mathcal{W}_1(\mathbf{p}_i(\tau), \ \mathbf{p}_j(\tau)) \le \epsilon'\}$$
$$= \min\{\tau : \ \kappa_{ij}(\tau) \ge 1 - \epsilon'\} ,$$

with $\epsilon' = d_0 \epsilon$ which is what we set out to show. Note that choosing any $\epsilon' \in (0, 1/2)$ ensures exponential convergence rate to the stationary measure[59] and by convention, we take the middle of this range and define $\tau_{ij}^{\mathrm{mix}} := \tau_{ij}^{\mathrm{mix}}(1/4)$ to obtain Eq. (3).

**Connection between geometric modularity and the symmetric stochastic block model**. In this section, we prove that the Boltzmann distribution of cluster assignments given the edge curvatures $\mathbb{P}(C|\boldsymbol{\kappa})$ (Eq. (5)) has equilibrium states which are indistinguishable from the ground truth partition of the SBM. We show this by reducing $\mathbb{P}(C|\boldsymbol{\kappa})$ as well as the posterior distribution $\mathbb{P}(C|G)$, to the same constant interaction Ising model (Eq. (7)). In the remainder of this section we work in the sparse regime, where $p_{\mathrm{in}}, p_{\mathrm{out}} = O(1/n)$.

First, we recap the well-known equivalence of the SBM and the Ising model[31]. Let $E$ denote the set of edges. The probability distribution of the symmetric SBM for two clusters can be written as[41]

$$\mathbb{P}(G|C) = p_{\mathrm{out}}^e (1 - p_{\mathrm{out}})^{\binom{n}{2} - e}$$
$$\times \prod_{ij \in E} \left(\frac{p_{\mathrm{in}}}{p_{\mathrm{out}}}\right)^{\delta(C_i, C_j)} \prod_{ij \notin E} \left(\frac{1 - p_{\mathrm{in}}}{1 - p_{\mathrm{out}}}\right)^{\delta(C_i, C_j)} \qquad (15)$$
$$\propto \prod_{ij \in E} \left(\frac{p_{\mathrm{in}}}{p_{\mathrm{out}}}\right)^{\delta(C_i, C_j)}$$

where $e$ is the total number of edges and in the last line we used that the effect of non-edges is weak in the sparse regime. Therefore, by Bayes' theorem with uniform prior one obtains the posterior distribution $\mathbb{P}(C|G) \propto \mathbb{P}(G|C)$. As a result, the probability of clusters generated by the SBM is equivalent to the Ising model with uniform interaction with Boltzmann distribution given by Eq. (7)[31].

Second, we reduce the Boltzmann distribution of clusters given the edge curvature to same Ising model in Eq. (7). From Eq. (5) we have

$$\mathbb{P}(C|\boldsymbol{\kappa}) \propto e^{\sum_{ij} \kappa_{ij}(\tau)\delta(C_i, C_j)}$$
$$\propto e^{\sum_{ij}[1 - \mathcal{W}_1(\mathbf{p}_i(\tau), \mathbf{p}_j(\tau))]\delta(C_i, C_j)} , \qquad (16)$$

where in the last line we used the definition of the curvature in Eq. (2). Comparing Eq. (16) with Eq. (7) note that $1 - \mathcal{W}_1(\mathbf{p}_i(\tau), \ \mathbf{p}_j(\tau))$ is non-constant and has a non-linear dependence on the scale $\tau$. However, it is possible to express it in terms of $p_{\mathrm{in}}, p_{\mathrm{in}}$ to make the connection to the Ising model. Let us write the diffusion

measures in Eq. (1) in terms of the spectral decomposition of $\mathbf{L}$ as

$$p_i^k(\tau) = \sum_{s=1}^n e^{-\lambda_s \tau} \phi_s(k) \phi_s(i) . \qquad (17)$$

At this point let us remark that in the dense regime where $p_{\mathrm{in}}, p_{\mathrm{out}} = O(1)$, the first two eigenmodes $(\lambda_1, \phi_1)$ and $(\lambda_c, \phi_c)$ dominate and the second eigenmode contains the anti-symmetric eigenvector $\phi_c(u) = 1/\sqrt{n}$ when $C_u = 1$ and $-1/\sqrt{n}$ when $C_u = 2$ that is associated with the community structure (Fig. 2c). Thus, one can follow spectral clustering methods[29] to find the sparsest cut between clusters using $\phi_c$. In contrast, in the sparse regime, the dominant eigenmodes will be driven by random fluctuations in the node degrees across the graph[60], thus spectral clustering algorithms based on $\mathbf{L}$ are suboptimal (Fig. 2d).

However, the coupled diffusion pair allows for cancelling out random fluctuations in their spectrum. To see this, consider for a between-edge $ij$ the difference

$$\sum_{ij \in E} p_i^k(\tau) - p_j^k(\tau) = \sum_{ij \in E} \sum_{s=1}^n e^{-\lambda_s \tau} \phi_s(k)[\phi_s(i) - \phi_s(j)]$$
$$= \sum_{s=1}^n e^{-\lambda_s \tau} \phi_s(k) \Delta \phi_s , \qquad (18)$$

where $\Delta \phi_s$ is defined in Eq. (8). The first term involves the constant eigenvector $\phi_1$ corresponding to the stationary state. Therefore, $\phi_1(i) = \phi_1(j)$ for all $ij$ and thus its contributions cancels out when taking differences. Further, for eigenvectors $\phi_s$ with $s \ne 1, c$ we have asymptotically ($n \to \infty$) that (Fig. 3)

$$\Delta \phi_s \to 0$$

As a result, the only contribution we are left with is coming from the anti-symmetric eigenmode $(\lambda_c, \phi_c)$. Thus we have that

$$\sum_{ij \in E} (p_i^u(\tau) - p_j^u(\tau)) = \begin{cases} \epsilon_\phi, & \text{if } C_i = C_j, \\ e^{-\lambda_c \tau} \phi_c \Delta \phi_c + \epsilon_\phi, & \text{if } C_i \ne C_j , \end{cases} \qquad (19)$$

where $\epsilon_\phi$ represents the contribution from the random eigenvectors which is negligible in the limit $n \to \infty$.

To compute $\mathcal{W}_1$ in the exponent of Eq. (16), we use Kantorovich–Rubinstein duality (Eq. (12)). Using Eq. (19) in Eq. (12) and ignoring asymptotically small terms, we consider the quantity

$$\sum_{ij \in E} \sum_u f(u)\left[p_i^u(\tau) - p_j^u(\tau)\right]$$
$$= \sum_{ij \in E} e^{-\lambda_c \tau} \sum_u f(u)\phi_c(u)$$
$$= \sum_{ij \in E} \frac{e^{-\lambda_c \tau}}{n}\left[\sum_{u:\, C_u = 1} f(u) - \sum_{u:\, C_u = 2} f(u)\right] \qquad (20)$$
$$= \sum_{ij \in E} \frac{e^{-\lambda_c \tau}}{n}\left[\sum_{u:\, C_u = 1}(f(u) - f(i)) - \sum_{u:\, C_u = 2}(f(u) - f(j))\right.$$
$$\left. + \sum_{u:\, C_u = 1} f(i) - \sum_{u:\, C_u = 2} f(j)\right].$$

In the sparse regime, we may make a tree-like approximation in the neighbourhood of $i$. This means that the number of neighbours of $i$ at distance $q$ inside the cluster is $p_{\mathrm{in}}^q (n/2)^q$, ignoring terms of order $O(1/n)$ and beyond. Considering only nodes at unit distance ($q = 1$), we approximate Eq. (20) as

$$\sum_{ij \in E} \frac{e^{-\lambda_c \tau}}{n}\left[\sum_{\substack{u:\, C_u = 1 \\ u \sim i}}(f(u) - f(i)) - \sum_{\substack{u:\, C_u = 2 \\ u \sim i}}(f(u) - f(i))\right.$$
$$+ \sum_{\substack{u:\, C_u = 1 \\ u \sim j}}(f(u) - f(j)) - \sum_{\substack{u:\, C_u = 2 \\ u \sim j}}(f(u) - f(j))$$
$$\left. + \sum_{\substack{u:\, C_u = 1 \\ u \sim i}} f(i) - \sum_{\substack{u:\, C_u = 2 \\ u \sim i}} f(i) + \sum_{\substack{u:\, C_u = 1 \\ u \sim j}} f(j) - \sum_{\substack{u:\, C_u = 2 \\ u \sim j}} f(j)\right]$$
$$= \sum_{ij \in E} \frac{e^{-\lambda_c \tau}}{n}\left[\sum_{\substack{u:\, C_u = 1 \\ u \sim i}}(f(u) - f(i)) - \sum_{\substack{u:\, C_u = 2 \\ u \sim i}}(f(u) - f(i))\right.$$
$$+ \sum_{\substack{u:\, C_u = 1 \\ u \sim j}}(f(u) - f(j)) - \sum_{\substack{u:\, C_u = 2 \\ u \sim j}}(f(u) - f(j))$$
$$\left. + \frac{n}{2}p_{\mathrm{in}}(f(i) - f(j)) - \frac{n}{2}p_{\mathrm{out}}(f(i) - f(j))\right].$$

Then, taking the supremum over all 1-Lipschitz functions $f$, we obtain

$$\sum_{ij \in E} \mathcal{W}_1(\mathbf{p}_i(\tau), \ \mathbf{p}_j(\tau))\delta(C_i, \ C_j)$$
$$\approx \sum_{ij \in E} e^{-\lambda_c \tau}(p_{\mathrm{in}} + p_{\mathrm{out}})\left(1 + \frac{|p_{\mathrm{in}} - p_{\mathrm{out}}|}{2(p_{\mathrm{in}} + p_{\mathrm{out}})}\right)\delta(C_i, C_j) \qquad (21)$$

Substituting this into Eq. (16) and noting that $p_{\mathrm{in}} + p_{\mathrm{out}}$ is constant we obtain at a

fixed $\tau$

$$\mathbb{P}(C|\kappa) \propto \exp\left[\left(\frac{|p_{in} - p_{out}|}{2(p_{in} + p_{out})}\right) \sum_{ij \in E} \delta(C_i, \ C_j)\right],$$

which up to a constant of proportionality equals the expression in Eq. (9).

## Data availability

The data generated in this study is available at https://dataverse.harvard.edu/dataverse/geometric_clustering/.

## Code availability

The code to reproduce the results in our paper and to perform geometric modularity optimisation is available at https://doi.org/10.5281/zenodo.5031276.

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

## Acknowledgements

A.G. acknowledges support from an HFSP Cross-disciplinary Postdoctoral Fellowship (LT000669/2020-C). This study was supported by funding to the Blue Brain Project, a research center of the École polytechnique fédérale de Lausanne (EPFL), from the Swiss government's ETH Board of the Swiss Federal Institutes of Technology. We thank Mauricio Barahona for insightful discussions on this topic, Jonas Braun and István Tomon for their helpful comments on the manuscript and Daniel Morales for inspiring us to analyse the *C. elegans* dataset. We also thank the three anonymous reviewers for their constructive comments.

## Author contributions

A.G. and A.A. contributed equally to this work.

## Competing interests

The authors declare no competing interests.
