## [Peer Review File · Nature Communications]

Reviewers' Comments:

Reviewer #1:

Remarks to the Author:

This paper considers the problem of hierarchical or multiscale community detection in a complex network.

The authors combine Ollivier Ricci curvature with a diffusion process. Consider a network and define a standard diffusion process at each node, then calculate the Ollivier Ricci curvature between two nodes i, j , which involves two distributions, taken as the distribution of the diffusion starting from i and j respectively, at time t .

Prior work has used Ollivier Ricci curvature for identifying bottleneck edges (vulnerable edges) and have used both Ollivier Ricci curvature itself and Ollivier Ricci curvature flow for community detection. The difference in this paper is to combine Ollivier Ricci curvature with a diffusion process.

This is an interesting idea. As a diffusion process would first propagate within a cluster, at different time t , one could observe clustering structures at multiple scales. The paper presented simulation results and some theoretical results with asymptotic behaviors in stochastic block models.

I find the idea to be interesting and of value. Finding hierarchical clusters is also an important problem. The paper is also generally well written.

There are places that the paper could be improved, in particular, in terms of comparison with prior community detection methods. The paper reported comparison with Markov stability and demonstrated superior performance. But given the vast amount of prior work on community detection, comparison with only one seems to fall short.

The previous method of using Ollivier Ricci flow can also detect hierarchical clusters. It would be very interesting to compare with that, to see what is the real benefit from using the distribution with diffusion for Ollivier Ricci curvature.

There are also many other methods for multi-scale community detection. For example:

Hierarchical Community Detection via Rank-2 Symmetric Nonnegative Matrix Factorization
<https://www.cc.gatech.edu/~hpark/papers/HierCommunity2017.pdf>

Graph Wavelets for Multiscale Community Mining
Nicolas Tremblay; Pierre Borgnat

I would encourage the authors to do a full literature search and compare with prior methods as well.

Minor issue:

Page 2, page 64, $A_{ij} = \max_{ij} w_{ij} - w_{ij}$. I am not sure I understand what this means.

Reviewer #2:

Remarks to the Author:

In this manuscript, the authors define a time varying edge curvature in order to study arbitrary networks, where the weights measure the similarity between pairs of dynamical network processes. Similar to some of the prior works, the authors show that the edge curvature distribution exhibits gaps at some time scales. The paper is in general well written, but there are few issues that require clarification or improvement:

1) One important question is whether the proposed Ollivier-Ricci curvature based edge clustering method is able to detect and distinguish community structure of nodes as spectral and node

clustering methods? For example, in the SBM network experiments, the in-cluster linking probability p_{in} are identical over different blocks/communities, therefore all in-community edges are equivalent (as in figure 2b, orange curves include all in-community edges even though there can be multiple communities; in figure 1f, there are three groups of edge curvatures and each represents an edge probability 0.8/0.1/0.02, but there can be more edge probabilities with nodes in different combination of the communities). However, what if the linking probability p_{in} varies with different communities (e.g., $p_{out}=0.05$, $p_{in}=0.5$ for cluster A and $p_{in}=0.4$ for cluster B)? Is it possible to differentiate the edges within cluster A and cluster B by examining the dynamic Ollivier-Ricci curvature of the edges?

2) Regarding lines [#64-69, Eq. 1], the definition of the probability measure of diffusion starting from unit mass δ_i is defined in Eq. 1 closely resembles the diffusion equation of a network. Trying to understand this conceptually, this is a distribution of the starting mass δ_i diffused across the (entire?) network structure (as encapsulated by the graph Laplacian) at different time scales τ but at a fixed rate of diffusion. Is this correct? Also, how different diffusion rates can be captured?

3) Following up my above comment, the inclusion of this rate of diffusion parameter could also provide some interesting addition to the proposed dynamical ORC formulation similar to the idleness parameter in the classical ORC. Can the authors provide additional clarifications?

4) The authors state "Here, instead of structural neighbourhoods we consider distributions generated by diffusion processes across scales τ ." My understanding is that instead of the fixed mass distribution on the adjacent neighbors assumed in the classical ORC formulation (Eq. 12), the dynamical ORC instead applies the probability measure of diffusion as defined in Eq. 1. Tying this with the previous question 2) and the analogy with the classical ORC, is the computation of the optimal transport distance (via the Wasserstein distance) now spread throughout the entire network or still limited to the immediate neighbors of the two end nodes? If it is the former, I can foresee time complexity issues for larger network sizes. The complexity analysis should be discussed similar to prior works cited in this work like [17].

5) The following statement "... the classical OR curvature measures the change of one-step neighbourhoods between nodes." is not entirely clear to me. I believe this statement is trying to relate the analogy of the mass transport (via the optimal transport theory) in a Riemannian manifold to a coarse geometry such as a network. Please provide additional explanations in the manuscript.

6) Understanding the extension of the classical ORC to the dynamic ORC intuitively as defined and described in Eqs. 1 & 2 is critical. Redefinition of the $p_i(\tau)$ generated by diffusion processes wrt the mass distributions p_i is not straight-forward. I would suggest to the authors to provide additional explanations of this analogy/extension between the classical and dynamic ORCs will give the paper more conceptual clarity.

7) Regarding the lines [#86-88] and Fig. 1a,b, it is not obvious to me how the scale organization is clearly seen from these subfigures. Please provide additional explanations and analysis.

8) I think in line [#95] "If i, j lie within the same subnetwork..." a subnetwork refers to a cluster, right? Please clarify.

9) In line [#113], "Fig. 1f shows three groups of edges, those with most positive curvature are found within clusters, while the other two groups of edges are found between pairs of clusters." I'm not clear which part of the figure is being referred. To me two groups of edges (those containing the colored lines) have positive dynamic ORCs across all time scales. Annotating the groups of edges in the figure could help.

10) Regarding line [#173] and Eq.5, I believe that the $\delta(C_i, C_j)$ here is the Kronecker delta. Please define. As the δ notation is also used to indicate the gap.

11) In lines [#226-229], the geometric modularity function definition is shown in Eq. 10. I believe the C_i and C_j are the cluster assignments. However, earlier in the text the C_i 's are defined as the ground truth in line #127. This could cause some confusion. In addition, main difference from the classical modularity maximization procedure is that the proposed geometric modularity is also a function of the time scale τ . Thus, the optimization procedure has to iterate across the time scales. Please clarify if a specific time scale is used for clusterization.

12) The authors mention in the introduction that the "...curvature provides a natural parametrization of complex networks to unveil their self-similar clusters across scales", yet the discussion does not refer to existing works that quantify the self-similarity of complex networks and determine their embedding dimensions. There have been several efforts concerning the embedding dimension such as Origins of fractality in the growth of complex networks. Nature Phys

(2006), Reliable Multi-Fractal Characterization of Weighted Complex Networks: Algorithms and Implications. Sci Rep (2017), Chimera states in complex networks: interplay of fractal topology and delay. Eur. Phys. J. Spec. Top. 226, 1883–1892 (2017), Chimera states in brain networks: Empirical neural vs. modular fractal connectivity Chaos 28, 045112 (2018). In general the complex network literature on self-similarity, embedding dimensions, etc. should be better discussed.

13) The authors mention in the abstract and discussion that the curvature and embedding dimension can be used to “tune the geometry of the graph to control the flux or interaction of network-driven dynamical processes”. This is indeed an important problem and several efforts are underway like Controlling the Multifractal Generating Measures of Complex Networks (2020).

14) The two case studies on the power grid and the gene expression are very interesting, but to me it seems that the authors missed to capitalize on the advantages of the method. How does the dynamic ORC compare to other community detection methods like modularity-based, eigenvalue based, previous time independent ORC methods? I can see a decrease in error from Fig 4.i but one can wonder if this is the best it can be achieved.

15) Lastly, the manuscript requires a careful reading to avoid hard to comprehend or grammar issues like “regime where when there is no...”. There are not many such issues, but a careful pass would eliminate even the ones I missed.

Overall, I think this is an interesting idea and hopefully my comments will help to improve the manuscript.

Reviewer #3:

Remarks to the Author:

This is the latex version of the review. I attached the pdf file below.

```
\documentclass[12pt]{article}
```

```
\begin{document}
```

```
\begin{center}
```

```
{\large {\bf Review of ``Unfolding the multiscale structure of networks with dynamical Ollivier-Ricci curvature'}}\end{center}
```

```
\bigskip \noindent
```

This paper is built around a multiscale extension of the Ollivier-Ricci (OR) curvature on weighted graphs.

In general, given a measure metric space X equipped with metric d_X and probability measures μ_x at each point $x \in X$, one defines the OR curvature $\kappa(x,y)$ as $\kappa(x,y) = 1 - \frac{W_1(\mu_x, \mu_y)}{d(x,y)}$. In the case of an (undirected) weighted graph (G,V,W) (V denotes the set of vertices and W denotes the set of positive weights w_{ij}), with distance d_{ij} between nodes i and j (e.g., the hop distance) one sets,

$\kappa_{ij} = 1 - \frac{W_1(\mu_i, \mu_j)}{d_{ij}}$, where

$\mu_i(j) = \frac{w_{ij}}{\sum_{k \sim i} w_{ik}}$.

Now the authors propose the following twist. Defining the normalized graph Laplacian L , they diffuse the measures μ_i via

```
\begin{equation} \label{eq:2} \mu_i(\tau) = \delta_i e^{-\tau L}, \end{equation}
```

and accordingly define a dynamic version of OR via

```
\begin{equation} \label{eq:1} \kappa_{ij}(\tau) = 1 - \frac{W_1(\mu_i(\tau), \mu_j(\tau))}{w_{ij}}. \end{equation}
```

Here $\delta_i(j) = 1$ for $i=j$ and 0 otherwise.

The intuition is to make OR multiscale. Note that at $\tau=0$, $\kappa_{ij}(0)$ will be 0, then when the measures diffuse to steady state π , one gets that $\kappa_{ij}(\tau)=1$. (There is a small typo on lines 102 and 103 where d_{ij} should be replaced by w_{ij} .) The bottom line, as the authors argue, is that the characteristic scales should be related to the overlap of pairs of diffused measures. This is used as a measure of information propagation on the various subnetworks. Indeed, they derive an upper bound on the mixing time of the diffusion pair. Finally,

the method is tested to derive some interesting results on several test cases, such as the European power grid and the *C. elegans* gene regulatory network.

The experiments look nice, and the paper seems technically correct. The authors do an excellent job analyzing their approach. My problem is ``impact.' There have been other diffusion based methods applied to such problems, e.g., in references [8] and [23]. In fact, even in the original paper of Ollivier [9] and in Bauer-Jost [14], diffused versions of the Ollivier-Ricci curvature are considered. So we have another proposed algorithm with a diffusion-based version of OR, which gives some intriguing and sensible results on some well-chosen examples.

This being said, I like the paper, it is very well-written, and I enjoyed reading it. Every method has its strengths and weaknesses, and it is not fair to demand an endless series of comparisons with other approaches. Nevertheless, I am curious about a comparison with another related methodology in which one uses a ``diffusion' type equation to evolve the metric via OR and detect hubs, subnetworks, communities, etc. Namely, one could use a Ricci flow. There have been several papers on this, but a well-explained one is ``Community Detection on Networks with Ricci Flow,' by

Chien-Chun Ni, Yu-Yao Lin, Feng Luo and Jie Gao, *Scientific Reports* volume 9, 2019. The code is available and easy to run (my group has done it). I think it would be very interesting to see how the discrete Ricci flow compares with the method proposed here. Some of the results seem similar. So in the present paper, one is diffusing the node based measures, and in discrete Ricci flow, via the OR curvature one is changing the metric.

A couple of other minor remarks: The OR curvature seems to be unnecessary and can be replaced by the ratio of the W_1 distance of the diffused measures (eq:2) and the weightings w_{ij} . On very large networks, the computation of e^{-tL} could be challenging even though there is a body of work treating this issue, e.g., due to the Andrea Bertozzi group. Have the authors thought of applying their method to partition (segmenting) images, in which one gets very dense large graphs?

Summary of revision actions

We thank the Reviewers for their insightful comments and the careful reading of our manuscript. Before addressing the Referee's comments point-by-point, here we list the major changes to the revised manuscript. We have also included a *diff* file to mark all changes.

Our additions have centred around the main criticisms of all Reviewers about the need for improved comparisons with other methods, particularly relating to the performance of our geometric clustering algorithm.

1) Performance comparison on generative graphs

We now perform a detailed accuracy comparison of our geometric modularity clustering algorithm against other methods on two widely-used benchmark graphs: the Stochastic Block Model (SBM) and the Lancichinetti-Fortunato-Radicchi (LFR). See Supplementary Note 3 and Supplementary Figure 2.

Since our paper focuses on the limits of detectability of clusters, we performed the SBM benchmark in the sparse regime (the average degree (3) was much less than the number of nodes (1000)). Clustering in this regime is particularly challenging because the clusters do not contain a 'core', which could be captured by density-based algorithms. Geometric modularity performed close to the theoretical (Kesten-Stigum) limit given by the belief propagation method (see Decelle et al., PRE, 2011 and Massoulié, STOC, 2014). As expected, geometric clustering outperformed classical node clustering methods including spectral clustering and Girvan-Newman edge betweenness (Supplementary Fig. 2a). Remarkably, geometric modularity also significantly outperformed classical modularity, which reiterates that it is the edge curvature distribution that encapsulates the cluster structure and not density differences, which could otherwise be captured by modularity alone.

As requested by all Reviewers, we also attempted to compare geometric modularity with the Ricci-flow method (Ni et al. Sci Rep, 2019), but found that it could not detect clusters in sparse graphs. Although Ni et al. did perform tests using the SBM benchmark, they did so in the dense regime (they fixed $p_{in}=0.15$ and varied p_{out}/p_{in} between 0.1 and 1, so the edge density they considered was at least $(0.1+0.15)500 = 125$, which is two orders of magnitude higher than 3 used in our study). This is in line with theoretical results showing that Ollivier-Ricci curvature is sensitive to local degree fluctuations (Proposition 2, Jost, Liu, Discrete Comput Geom, 2013). This reinforces our theoretical insight that combining diffusion processes (constructed from the graph Laplacian) and OR curvature is what allows suppressing fluctuations in the Laplacian spectrum (Fig. 3) thus bypassing the inherent limitations of previous methods.

We also performed a benchmarking on the LFR generative network, where community sizes and edge densities are heterogeneous (see comment 1 of Reviewer 2). We show that our method remains robust performing comparably to the state-of-the-art spinglass and modularity

methods (Supplementary Fig. 2b). In this uniscale benchmark, the fact that geometric modularity performs as well as standard modularity is a statement that information is not lost by reweighting the edges by the curvatures and that we indeed identify the correct clustering scale. Here we could also compare to the Ricci flow method of Ni et al., showing that our method performs substantially better.

2) Performance comparison on the *C. elegans* homeobox gene network

The Reviewers also requested a better comparison with multiscale methods and diffusion-based methods. To emphasize that our method is not specifically designed to perform on generative networks, we decided to use the *C. elegans* homeobox gene network (Figure 4) as a real-world graph example, which has the added benefit that the ground truth is known (see Supplementary Node 4 and Supplementary Fig. 3). We compared against a range of well-known methods, including several based on diffusions, modularity, Laplacian spectrum and matrix-factorisation. An important feature of any good multiscale methods is to robustly detect clusters at meaningful scales, while rejecting other scales. We found that most methods did not have this feature and either over-partition the graph or fail to resolve clusters at the small resolution. In contrast, we found that geometric modularity performed better than the wide range of multiscale and uniscale methods we studied. Our method returned a large plateau of meaningful scales, which are all very close to the ground truth and vary only by aggregating neighbouring small clusters (Fig. 4g), demonstrating a high degree of robustness to noise. The method of Tremblay and Borgnat, IEEE, 2014 (suggested by Reviewer 1) also performed remarkably well and the best returned cluster (of lowest VI) was close to the ground truth.

3) Clarifying conceptual differences

We clarified conceptual differences of the multiscale geometric notion developed in our work with other diffusion-based approaches and with other Ollivier-Ricci curvature-based approaches with particular focus on the recent Ricci-flow method of Ni et al, 2019, as requested by all Reviewers. We also extended the Introduction and Discussion to present a more comprehensive literature review of the relevant methods as suggested by the Referees.

Overall, we thank the Referees for their thoughtful critiques and feel that these revisions have significantly clarified and strengthened our contributions.

Reviewer 1

This paper considers the problem of hierarchical or multiscale community detection in a complex network.

The authors combine Ollivier Ricci curvature with a diffusion process. Consider a network and define a standard diffusion process at each node, then calculate the Ollivier Ricci curvature between two nodes i, j , which involves two distributions, taken as the distribution of the diffusion starting from i and j respectively, at time t .

Prior work has used Ollivier Ricci curvature for identifying bottleneck edges (vulnerable edges) and have used both Ollivier Ricci curvature itself and Ollivier Ricci curvature flow for community detection. The difference in this paper is to combine Ollivier Ricci curvature with a diffusion process.

This is an interesting idea. As a diffusion process would first propagate within a cluster, at different times t , one could observe clustering structures at multiple scales. The paper presented simulation results and some theoretical results with asymptotic behaviors in stochastic block models.

I find the idea to be interesting and of value. Finding hierarchical clusters is also an important problem. The paper is also generally well written.

We thank the Reviewer for the enthusiastic evaluation of our manuscript.

There are places that the paper could be improved, in particular, in terms of comparison with prior community detection methods. The paper reported comparison with Markov stability and demonstrated superior performance. But given the vast amount of prior work on community detection, comparison with only one seems to fall short.

In response to this comment, we performed detailed comparisons on two benchmark graphs (SBM and LFR) with the state-of-the-art methods on these benchmarks as well as further comparisons on the *C. elegans* homeobox gene network. Please see points 1-2 on the introductory page for details and see Supplementary Notes 3,4 and Supplementary Figs. 2,3 for the results.

In addition, we extended our literature review and more explicitly clarified the connection to related classes of methods.

The previous method of using Ollivier Ricci flow can also detect hierarchical clusters. It would be very interesting to compare with that, to see what is the real benefit from using the distribution with diffusion for Ollivier Ricci curvature.

We believe that the Reviewer refers to the work of Ni et al. Sci Rep, 2019, which we cited in the original manuscript (Ref. 18). This method detects communities in a two-step heuristic. In step 1, it uses Ricci flow to evolve the edge weights until convergence, i.e., the curvature (in the classical Ollivier-Ricci sense) is as flat as possible. During this process, densely connected nodes will get closer (their edge weight decreases) while sparsely connected nodes move farther. In the second step, they use thresholding to cut edges that are further than a predefined length.

We now show that our method substantially outperforms the Ricci flow method on the LFR benchmark and, unlike the Ricci flow method, it can detect clusters in sparse graphs where the detection problem is particularly hard. See summary point 2 on the introductory page above along with Supplementary Note 3 and Supplementary Figure 2.

We have not performed further comparisons on our multiscale examples because of fundamental differences and limitations of the Ricci flow method.

Ricci flow method (Ni et al.) – Not a true multiscale method because they evolve the Ricci flow until convergence. Thus the multiscale nature is discarded as the algorithm is asked to output only one geometric representation (edge weight distribution after converged Ricci flow).

>> Our method – Diffusions provide different geometric representations at certain characteristic time scales (see next point on how we define these scales).

Ricci flow method (Ni et al.) – In one of their use cases (Figure 7 in Ni et al., 2019) they claim to detect hierarchical clusters by choosing different cutoff thresholds. However, their chosen cutoff points are selected by hand and they provide no comparison to a ground truth or justification that they in fact find meaningful communities.

>> Our method – We found conditions when to stop the diffusions to obtain informative curvature distributions: based on maximal curvature gap (Fig. 2e,f) and based on minima of variation of information (Fig. 4). Both of these conditions are theoretically justified: maximising curvature gap optimally detects communities in SBMs (Fig. 2h, i), whereas minimising VI is motivated by the fact that we search for equilibrium solutions of the Boltzmann distribution induced by the edge curvatures (Eq. (5)).

We rewrote the paper at several points to highlight the benefits for using diffusion processes in the construction of the edge curvature.

There are also many other methods for multi-scale community detection. For example:

Hierarchical Community Detection via Rank-2 Symmetric Nonnegative Matrix Factorization

<https://www.cc.gatech.edu/~hpark/papers/HierCommunity2017.pdf>

Graph Wavelets for Multiscale Community Mining

Nicolas Tremblay; Pierre Borgnat

We thank the Reviewer for the suggestion. To address this point, we re-analyzed the *C. elegans* example against a host of other methods including the wavelet method by Tremblay and Borgnat (see points 2 in the summary above along with Supplementary Note 3 and Supplementary Fig. 3). We found that most methods underfit or overfit the cluster structure. Our method and the wavelet method were among the few methods that captured the correct clustering scale.

We could not test our method against the `hiernmf2` method recommended by the Reviewer. The code corresponding to the paper is designed to cluster data points with feature vectors and not graphs. So although it could apply to our *C. elegans* dataset, this would be not a fair comparison with graph clustering methods.

I would encourage the authors to do a full literature search and compare with prior methods as well.

Thank you for this feedback. We have now extended our literature review in the Introduction.

Minor issue:

Page 2, page 64, $A_{ij} = \max_{ij} w_{ij} - w_{ij}$. I am not sure I understand what this means.

We apologise for the confusion on this subtle but important point.

Perhaps the confusion is caused by our inaccurate use of subscripts. We have changed the expression to $A_{ij} = \max_{kl} w_{kl} - w_{ij}$.

Let us also clarify that to construct the diffusion process on the graph (Eq. 1) we need to interpret the entries in the adjacency matrix A_{ij} as node-to-node similarities (or affinities). Then the transition probabilities for the diffusion process are obtained by normalising these similarities by the node degree. However, when we compute the geodesic distance matrix (used in the computation of the optional transport distance W in Eq. 2) we need to define distances. Therefore, we decided to define edge weights as distances w_{ij} and construct the adjacency matrix as $A_{ij} = e^{-\{w_{ij}\}}$ or $A_{ij} = \max_{kl} w_{kl} - w_{ij}$. The latter is just a first order expansion of the former, while keeping all edge similarities positive.

We have further clarified this in the revised manuscript.

Reviewer 2

In this manuscript, the authors define a time varying edge curvature in order to study arbitrary networks, where the weights measure the similarity between pairs of dynamical network processes. Similar to some of the prior works, the authors show that the edge curvature distribution exhibits gaps at some time scales.

We thank the Reviewer for the summary of our work. Let us, however, emphasise that our *time-dependent edge curvature* (Eq. (1)-(2)), and the resulting ‘*curvature gap*’ (Eq. (4)) are both important novel concepts of our paper. We agree that other (time-indepenent) edge curvature notions may also heuristically lead to curvature gaps when strong communities are present. However, we go well beyond heuristic methods by providing solid theoretical support to the concept of curvature gap.

1. The curvature gap encodes the presence of edges in the graph with limited information flow. Edges constituting the curvature gap are bottleneck edges across which mixing is incomplete relative to the clusters to which the endpoints of these edges belong (Fig. 1, Eq. (3)).
2. The curvature gap is a robust extension of the ‘spectral gap’ that exists until the fundamental limit of detecting communities (Fig. 2h, i).
3. Since curvatures optimally encode communities we could derive the geometric modularity algorithm, which therefore does not require additional parameters or a statistical null model.

As also requested by Reviewer 1, we expand the literature background of our work and take this opportunity to emphasise the important advances 1-3 in our work.

The paper is in general well written, but there are few issues that require clarification or improvement:

- 1) One important question is whether the proposed Ollivier-Ricci curvature based edge clustering method is able to detect and distinguish community structure of nodes as spectral and node clustering methods?

This is an important question. In our paper, we put particular emphasis on clustering sparse graphs. This is partly because sparse SBM graphs are particularly hard to cluster as they do not enjoy community properties such as small world or preferential attachment and are locally tree-like with long cycles meaning that many density-based algorithms will fail.

We have now performed benchmarking against common clustering methods in two well-known graph classes (see point 2 on the introductory page, as well as Supplementary Note 3 and Supplementary Figure 2). We also compared geometric modularity to other methods on the *C. elegans* dataset (Supplementary Note 4, Supplementary Fig. 3).

For example, in the SBM network experiments, the in-cluster linking probability p_{in} are identical over different blocks/communities, therefore all in-community edges are equivalent (as in figure 2b, orange curves include all in-community edges even though there can be multiple communities; in figure 1f, there are three groups of edge curvatures and each represents an edge probability 0.8/0.1/0.02, but there can be more edge probabilities with nodes in different combination of the communities). However, what if the linking probability p_{in} varies with different communities (e.g., $p_{out}=0.05$, $p_{in}=0.5$ for cluster A and $p_{in}=0.4$ for cluster B)? Is it possible to differentiate the edges within cluster A and cluster B by examining the dynamic Ollivier-Ricci curvature of the edges?

We thank the Reviewer for this question. Indeed, in all stochastic block model examples in our paper we used equal cluster sizes and equal within-cluster edge probabilities (p_{in}). In Fig. 1 this serves no particular purpose beyond simply being the simplest example to illustrate how to geometrically reveal characteristic scales in a multiscale graph.

In the revised manuscript, we changed Fig. 1 to having asymmetric communities of sizes (30,40,35,50) and corresponding p_{in} of (0.7,0.8,0.9,0.5). We kept the between community probabilities the same, 0.1 or 0.02. For illustration purposes, these need to be around an order of magnitude apart, otherwise the curvature gaps start to be visually harder to distinguish. However, small differences are still picked up by our curvature gap metric (see Fig. 2f as an example). This is the basis of our geometric modularity clustering algorithm demonstrated in Fig. 4.

To show that the algorithm works for asymmetric clusters we have performed a comparison with the LFR benchmark, in which both the within cluster probabilities and cluster sizes are drawn from exponential distributions and the number of clusters is also unknown (Supplementary Note 3 and Supplementary Fig. 2b). This benchmark is uni-scale because the between-cluster edge probabilities are set by the same mixing parameter μ_t . Our method performs at an accuracy similar to the state-of-the-art spinglass and modularity optimisation methods, yet still having the possibility to adjust the scale parameter in other multiscale scenarios (see also point 2 in the introductory page).

It is also important to note, in the SBM example (Fig. 2, 3), having equal community sizes and p_{in} is an important symmetry requirement, which is necessary to compare to known theoretical results. The algorithm does not assume equal cluster sizes and p_{ins} , however, currently, the theoretical results we refer to (in particular Refs [19-21,34]) are only proved for this case.

2) Regarding lines [64-69, Eq. 1], the definition of the probability measure of diffusion starting from unit mass δ_i is defined in Eq. 1 closely resembles the diffusion equation of a network. Trying to understand this conceptually, this is a distribution of the starting mass δ_i diffused across the (entire?) network structure (as encapsulated by the graph Laplacian) at different time scales τ but at a fixed rate of diffusion. Is this correct?

This is correct. Eq. 1 is the solution of a diffusion equation

$$dp_i/d\tau = -Lp_i$$

started from unit point mass δ_i . The masses, in theory, are supported on all nodes on the network and are preserved across τ (however, we explain in point 4 below how we 'trim' these distributions to reduce the complexity of our algorithm).

Also, how different diffusion rates can be captured?

In our current framework, we use diffusions to capture the features of the graph. Therefore, we consider the diffusion rates are proportional to the edge similarities and are fully captured in the normalised Laplacian. In effect, diffusion is the continuous-time analogue of an unbiased random walker moving from node to node on the graph with probability of moving from node i to j given by A_{ij} / K_i , where A_{ij} is the corresponding entry in the adjacency matrix and K_i is the degree of node i .

Varying diffusion rates could be captured by reweighting graph edges, i.e., considering biased diffusions on the graph. While varying the diffusion rates is an interesting extension of our framework, the systematic exploration of this direction is not within the scope of this work.

3) Following up my above comment, the inclusion of this rate of diffusion parameter could also provide some interesting addition to the proposed dynamical ORC formulation similar to the idleness parameter in the classical ORC. Can the authors provide additional clarifications?

The idleness parameter is important only for the classical ORC because of how the measures are defined: one-step random walk applied on a delta measure. In this *discrete* random walk, at $t=0$ all mass is on nodes i and j , whereas at time $t=1$ all mass is on the neighbours. The idleness parameter interpolates between these two extremes adjusting how much weight is given to the direct connection between adjacent nodes relative to the (transport) distance between their neighbours.

In our framework, the role of the idleness parameter is fully accounted for in the time (τ) parameter in the diffusion processes. Moreover, as mentioned in the response of the previous question, there is no rate of diffusion parameter in our formulation; the diffusion rates are fully encoded in the network weights. In this context, laziness corresponds to having a graph with self-loops, which, for us, corresponds to a choice of network, not a parameter in our notion of curvature.

We clarified this in the revised manuscript.

4) The authors state “Here, instead of structural neighbourhoods we consider distributions generated by diffusion processes across scales τ .” My understanding is that instead of the fixed mass distribution on the adjacent neighbors assumed in the classical ORC formulation (Eq. 12), the dynamical ORC instead applies the probability measure of diffusion as defined in Eq. 1. Tying this with the previous question 2) and the analogy with the classical ORC, is the computation of the optimal transport distance (via the Wasserstein distance) now spread throughout the entire network or still limited to the immediate neighbors of the two end nodes? If it is the former, I can foresee time complexity issues for larger network sizes. The complexity analysis should be discussed similar to prior works cited in this work like [17].

We thank the Reviewer for this question which was also raised by Reviewer 3. The computation of the dynamic OR curvature is moderately expensive and scales similarly to several other algorithms (random walk based algorithms, Girvan-Newman, belief propagation). We could practically use it to cluster graphs up to 10^4 nodes on a desktop computer. However, it is highly parallelizable and several approximative algorithms exist to increase the applicability of our method

In the revised work, we have expanded our discussion about the computational complexity and implemented several techniques for speed-up including see Supplementary Note 5 and Supplementary Fig. 4)

1. a cutoff to trim the measures, setting very small node values to zero, to speed up the computation of the optimal transport distance,
2. the Sinkhorn algorithm, based on entropy regularised optimal transport distance, to estimate the optimal transport distance,
3. GPU implementation of the Sinkhorn algorithm to parallelise the computation of the curvatures on the whole graph.

In addition, we expect complexity to dramatically reduce in the future with machine learning techniques to approximate optimal transport distances (c.f. Arjovski et al., arXiv, 2017).

5) The following statement “... the classical OR curvature measures the change of one-step neighbourhoods between nodes.” is not entirely clear to me. I believe this statement is trying to relate the analogy of the mass transport (via the optimal transport theory) in a Riemannian manifold to a coarse geometry such as a network. Please provide additional explanations in the manuscript.

We apologise for the lack of clarity here. We have improved this sentence in the text, which was meant to only be a brief, intuitive description of the original OR curvature.

6) Understanding the extension of the classical ORC to the dynamic ORC intuitively as defined and described in Eqs. 1 & 2 is critical. Redefinition of the $p_i(\tau)$ generated by diffusion processes wrt the mass distributions p_i is not straight-forward. I would suggest to the authors to provide additional explanations of this analogy/extension between the classical and dynamic ORCs will give the paper more conceptual clarity.

As mentioned in point 3) above, the measures defined in the original ORC can be viewed as a one step lazy (or with self-loops) discrete time random walk. Our dynamic ORC is a natural extension to more than one step, or to continuous time random, which provides an intrinsic timescale to the curvature.

We have elaborated on this analogy in the revised manuscript.

7) Regarding the lines [86-88] and Fig. 1a,b, it is not obvious to me how the scale organization is clearly seen from these subfigures. Please provide additional explanations and analysis.

We apologise for the confusion. Multi-scale structure in undirected graphs can emerge either due to large differences in edge densities or cluster sizes between different pairs of clusters. In Fig. 1, for illustration we fixed the cluster sizes (we relaxed this in the

revised manuscript) and introduced between-cluster edge densities that are approximately an order of magnitude different (0.02 and 0.1 in the original manuscript). Note that in the revised manuscript we introduced variations in the cluster sizes and between cluster probabilities in response to remark 1).

We have more carefully explained the construction of this graph in the paper.

8) I think in line [#95] “If i, j lie within the same subnetwork...” a subnetwork refers to a cluster, right? Please clarify.

Yes, we meant cluster. We clarified this.

9) In line [#113], “Fig. 1f shows three groups of edges, those with most positive curvature are found within clusters, while the other two groups of edges are found between pairs of clusters.” I’m not clear which part of the figure is being referred. To me two groups of edges (those containing the colored lines) have positive dynamic ORCs across all time scales. Annotating the groups of edges in the figure could help.

We thank the Reviewer for pointing this out. First, we would like to note that the relevant feature of the curvature evolution is the relative magnitude of edge curvatures at a given snapshot in time and not the sign of the curvature. In Fig. 1f we aimed to distinguish the three edge curvature ‘bundles’, which correspond to edges connecting regions with similar degree of connectivity. Specifically, one of the bundles (with the highest curvature) corresponds to within-cluster edges and the two other bundles correspond to between-cluster edges at the two scales.

We have now added insets to Fig. 1f to illustrate the position of the edges in the corresponding bundles.

10) Regarding line [#173] and Eq.5, I believe that the $\delta(C_i, C_j)$ here is the Kronecker delta. Please define. As the δ notation is also used to indicate the gap.

We have defined the Kronecker delta after Eq. (5) and changed the notation of the curvature gap to $\Delta\kappa(\tau)$.

11) In lines [#226-229], the geometric modularity function definition is shown in Eq. 10. I believe the C_i and C_j are the cluster assignments. However, earlier in the text the C_i 's are defined as the ground truth in line #127. This could cause some confusion.

Thank you for pointing out this inconsistency. We have updated the notation in the text to clarify the cluster assignments (C) vs the ground truth (C^*).

In addition, main difference from the classical modularity maximization procedure is that the proposed geometric modularity is also a function of the time scale τ . Thus, the optimization procedure has to iterate across the time scales. Please clarify if a specific time scale is used for clusterization.

To find the clustering scale, we run, at each scale, the Louvain algorithm on the curvature-weighted graph many (100) times with random initializations, and from the robustness of Louvain partitions via the variation of information we can assess candidate scales. In addition, similar clusters found across a large range of scales is another indication of a meaningful community structure.

12) The authors mention in the introduction that the “...curvature provides a natural parametrization of complex networks to unveil their self-similar clusters across scales”, yet the discussion does not refer to existing works that quantify the self-similarity of complex networks and determine their embedding dimensions. There have been several efforts concerning the embedding dimension such as Origins of fractality in the growth of complex networks. Nature Phys (2006), Reliable Multi-Fractal Characterization of Weighted Complex Networks: Algorithms and Implications. Sci Rep (2017), Chimera states in complex networks: interplay of fractal topology and delay. Eur. Phys. J. Spec. Top. 226, 1883–1892 (2017), Chimera states in brain networks: Empirical neural vs. modular fractal connectivity Chaos 28, 045112 (2018). In general the complex network literature on self-similarity, embedding dimensions, etc. should be better discussed.

Connections to fractal geometry is a very interesting research direction, which we have thought about pursuing. In fact, by stopping the diffusion processes at appropriate times and aggregating nodes one can design various coarse graining schemes based on our diffusive geometry which we think are related to renormalisation groups in fractal geometry. However, this connection is not immediate but one which we will likely pursue in the future. We have mentioned this research direction in the Discussion along with the references suggested by the Reviewer.

We must, however, clarify that we use Refs [3-4] as examples when latent space embeddings have provided insight into the multiscale organisation of an important class of networks (complex networks). However, this is in contrast to our formalism, which departs from the (often limiting) network embedding viewpoint and develops a geometric notion that applies to a general class of networks. As we write, “thus, there is

a need for a geometric notion that does not require embedding, yet allows studying the multiscale structure of a general class of networks”. Specifically, rather than focusing on characterising the embedding space, we focused on constructing a geometric object that has the expected behaviour in various limits (Supplementary Figure 1), and provides insight to limits of clustering on SBM graphs and multiscale clustering of real-world graphs. Therefore, we also focused the text in the Introduction to better distinguish our contribution from approaches that rely on embeddings.

13) The authors mention in the abstract and discussion that the curvature and embedding dimension can be used to “tune the geometry of the graph to control the flux or interaction of network-driven dynamical processes”. This is indeed an important problem and several efforts are underway like Controlling the Multifractal Generating Measures of Complex Networks (2020).

We agree that this is a research direction that our work will likely have the largest impact on. Motivated by this comment we elaborated in the Discussion on possible research directions concerning the study of synchronisation problems and chimera states by identifying information limiting edges.

14) The two case studies on the power grid and the gene expression are very interesting, but to me it seems that the authors missed to capitalize on the advantages of the method. How does the dynamic ORC compare to other community detection methods like modularity-based, eigenvalue based, previous time independent ORC methods? I can see a decrease in error from Fig 4.i but one can wonder if this is the best it can be achieved.

We thank the Reviewer for pointing out this shortcoming. We now include a detailed benchmarking of our method against other classical methods (including modularity-based, eigenvalues, based, and the time-independent ORC method of Ni et al, 2019) on generative graphs (SBM and LFR). See point 2 in the introductory page as well as Supplementary Notes 3,4 and Supplementary Figures 2,3. We find that our method performs close to the theoretical limit in both cases matching or outperforming state-of-the-art methods. Notably, our method does substantially better than the classical (time-independent) ORC method of Ni et al. We also performed comparison of our methods to a broad selection of other methods on the *C. elegans* network. We find that while most other methods either overfit or underfit the clusters, our method was among the few methods that could identify the correct clustering scale.

15) Lastly, the manuscript requires a careful reading to avoid hard to comprehend or grammar issues like “regime where when there is no...”. There are not many such issues, but a careful pass would eliminate even the ones I missed.

Thank you for this comment. We did a careful pass on the paper to remove these issues.

Overall, I think this is an interesting idea and hopefully my comments will help to improve the manuscript.

We thank the Reviewer for the enthusiastic evaluation of our work!

Reviewer 3

This paper is built around a multiscale extension of the Olliver-Ricci (OR) curvature on weighted graphs. In general, given a measure metric space X equipped with metric d_X and probability measures μ_x at each point $x \in X$, one defines the OR curvature $\kappa(x, y)$ as

$$\kappa(x, y) = 1 - W_1(\mu_x, \mu_y) / d(x, y)$$

In the case of an (undirected) weighted graph (G, V, W) (V denotes the set of vertices and W denotes the set of positive weights w_{ij}), with distance d_{ij} between nodes i and j (e.g., the hop distance) one sets,

$$k_{ij} = W_1(\mu_i, \mu_j) / d_{ij}$$

where

$$\mu_i(j) = w_{ij} / \sum_{k \sim i} w_{ik}$$

Now the authors propose the following twist. Defining the normalized graph Laplacian L , they diffuse the measures μ_i via

$$\mu_i(\tau) = \delta_i e^{-\tau L},$$

and accordingly define a dynamic version of OR via

$$k_{ij}(\tau) = 1 - W_1(\mu_i(\tau), \mu_j(\tau)) / w_{ij}$$

Here $\delta_i(j) = 1$ for $i = j$ and 0 otherwise.

The intuition is to make OR multiscale. Note that at $\tau = 0$, $k_{ij}(0)$ will be 0, then when the measures diffuse to steady state π , one gets that $k_{ij}(\tau) = 1$. (There is a small typo on lines 102 and 103 where d_{ij} should be replaced by w_{ij} .)

Thank you for pointing out the typo. We have corrected this along with a few other typos we have spotted.

The bottom line, as the authors argue, is that the characteristic scales should be related to the overlap of pairs of diffused measures. This is used as a measure of information propagation on the various subnetworks. Indeed, they derive an upper bound on the mixing time of the diffusion pair. Finally, the method is tested to derive some interesting

results on several test cases, such as the European power grid and the *C. elegans* gene regulatory network.

We thank the Reviewer for the summary of our manuscript.

The experiments look nice, and the paper seems technically correct. The authors do an excellent job analyzing their approach. My problem is “impact.” There have been other diffusion based methods applied to such problems, e.g., in references [8] and [23]. In fact, even in the original paper of Ollivier [9] and in Bauer-Jost [14], diffused versions of the Ollivier-Ricci curvature are considered. So we have another proposed algorithm with a diffusion-based version of OR, which gives some intriguing and sensible results on some well-chosen examples.

Indeed, we are not the first (nor the last) to use diffusion to study network structure. Diffusions (heat kernels) are central objects in the graph learning and graph signal processing literatures. However, we believe our work is the first that combines diffusions and discrete geometry to

- 1) Show that pairs of diffusions used in the construction of the curvature pick up random variation in the graph independently and allow uninformative fluctuations to be ‘averaged out’. In this light, the main difference between Refs. [8, 23] and our approach is that while those approaches (and other approaches relying on single diffusions) rely on the spectrum of the Laplacian being well-behaved (the eigenvalues decay sufficiently quickly), we show that pairs of diffusions can pick up structure in the graph well below what is expected from methods based on spectral properties of the Laplacian (see Figure 3).
- 2) As a result of the construction in point 1) our dynamic OR curvature captures the cluster structure near the fundamental limit of detection (Figure 2h, i). In contrast, previous ‘local’ OR curvature constructions are sensitive to degree variations. For example, in the revised work we show that the clustering method of Ni et al. (2019) fails to detect clusters when the graph is sparse (and hence node degree fluctuations are large). See next response for details.
- 3) Introduce a scale parameter in the definition of the curvature and show how to set this to obtain meaningful geometric representations and inspire a multiscale clustering algorithm (Figure 4). In contrast, Refs. [9, 14] have a different notion of diffusion OR curvature, which they define as the limit for $t \rightarrow 0$ of a time derivative term, and have no resulting time scale parameters.

This being said, I like the paper, it is very well-written, and I enjoyed reading it. Every method has its strengths and weaknesses, and it is not fair to demand an endless series of comparisons with other approaches. Nevertheless, I am curious about a comparison with another related methodology in which one uses a “diffusion” type

equation to evolve the metric via OR and detect hubs, subnetworks, communities, etc. Namely, one could use a Ricci flow. There have been several papers on this, but a well-explained one is “Community Detection on Networks with Ricci Flow,” by Chien-Chun Ni, Yu-Yao Lin, Feng Luo and Jie Gao, Scientific Reports volume 9, 2019. The code is available and easy to run (my group has done it). I think it would be very interesting to see how the discrete Ricci flow compares with the method proposed here. Some of the results seem similar. So in the present paper, one is diffusing the node based measures, and in discrete Ricci flow, via the OR curvature one is changing the metric.

This is a very interesting question that was also raised by other reviewers. As a result, we now discuss the comparison with the Ricci flow method of Ni et al.

In the Ricci flow method the edge weights w_{ij} are evolved according to

$$dw_{ij}/dt = -\kappa_{ij} w_{ij}$$

Where κ_{ij} is the classical Ollivier-Ricci curvature. Importantly, they run this process until they are within an epsilon distance from the converged state and, consequently, the time parameter does not play the role of a resolution parameter. Although it is expected that curvature evolution integrates information from the graph due to the diffusive nature of the Ricci flow, this information is aggregated into one geometric representation in the form of a set of edge weights $\{w_{ij}^*\}$ obtained by the converged Ricci flow. Instead, in our dynamical Ollivier-Ricci formulation, we evolve the diffusions, and in turn the edge curvatures, and stop this evolution at certain well-defined timescales. Therefore we decompose the graph into geometric representations containing features of increasing sizes. We use these representations to perform clustering. We kindly refer the Reviewer to our response to Reviewer 1 (on page 4 of this document), where we also outline several fundamental differences from the Ricci flow method of Ni et al.

We also provide performance comparisons with the Ricci flow method on two benchmark graphs (see also point 1 in the revision summary on the first page). Our method significantly outperformed the Ricci flow on the LFR benchmark. On the SBM benchmark, our method approached the theoretical limit of detection, whereas the Ricci flow method altogether failed to detect communities in the sparse regime.

Based on our previous experiments with Ricci flows we think it could be interesting in the future to combine the Ricci flow approach with Ni et al. with our dynamic Ollivier Ricci framework. At its simplest, picking a diffusive scale τ and running the Ricci flow until convergence could, in principle, enable the algorithm to extract features at different resolutions. However, at present this is very computationally expensive to perform.

A couple of other minor remarks: The OR curvature seems to be unnecessary and can be replaced by the ratio of the W1 distance of the diffused measures (1) and the weightings w_{ij} .

This is correct. However, we prefer to keep the constant part of the expression in order to preserve the intuition in canonical graph topologies such as trees, grids and cliques, yielding negative, zero and positive curvature, respectively (see Supplementary Figure 1).

On very large networks, the computation of e-tL could be challenging even though there is a body of work treating this issue, e.g., due to the Andrea Bertozzi group.

We refer to some replies to Reviewer 2 regarding possible improvements of the numerical complexity, but indeed, the computation of the matrix exponential is always the bottleneck in any continuous diffusion-based method. We use the scaling and squaring algorithm available in Python, as it is fast enough for the size of graphs we considered in this paper, but any other improvements on it are indeed welcome. We are not familiar with Andrea Bertozzi's work on that topic, nor could find it in her papers, but we will definitely keep it in mind for future improvement of our Python package to extend to larger graphs. Thank you for pointing this out.

Have the authors thought of applying their method to partition (segmenting) images, in which one gets very dense large graphs?

This is an interesting application, which could be a future research direction, but which we did not think about. We think our method will be most valuable to cluster sparse graphs where the limited number of edges conveys very little information about the cluster structure. Thus, learning a geometric representation of the graph as a first step can increase effectiveness of clustering algorithms. In the case of dense graphs, it may be relevant to first apply some sparsification algorithm in order to extract the 'backbone' structure of the graph from the 'noise' induced by its high density. We used this approach in the *C. elegans* example (Fig. 4g-j).

Reviewers' Comments:

Reviewer #1:

None

Reviewer #2:

Remarks to the Author:

I read the revised manuscript and the response letter and the changes made by the authors are substantial and commendable. I summarize below a few minor issues:

- In this response "In addition, we extended our literature review and more explicitly clarified the connection to related classes of methods.", I would have liked to see how the discussion in the manuscript has been enriched. Similar comment applies for this reviewer's comment "I would encourage the authors to do a full literature search and compare with prior methods as well." and response from the authors "Thank you for this feedback. We have now extended our literature review in the Introduction." where it would have been useful to learn what is the newly added information and how it changed the manuscript. Similarly for this comment of reviewer 2 "13) The authors mention in the abstract and discussion that the curvature and embedding dimension can be used to ..." for which the authors responded " We agree that this is a research direction that our work will likely have the largest impact on. Motivated by this comment we elaborated in the Discussion on possible research directions concerning the study of synchronisation problems and chimera states by identifying information limiting edges." Of note, the control or tuning of the network either by adding or removing links or by adjusting the edge weights goes beyond synchronization and can deal with controllability of complex networks. I believe that their proposed work can be well aligned with recent efforts on controlling multifractality of networks but this needs to be carefully discussed. In general, changes in revisions should have been highlighted. Please note that reference [28] is cited, but it is unclear where the comparison exists.
- I appreciate the comparison with the graph wavelet and the other method suggested by the reviewer, it would be nice if the codes are made available. Please explain where readers can find the code.
- Regarding comment 4 of reviewer 2 on computational complexity, the response doesn't shed light on the computational complexity, I assume it is bounded by the computational complexity of the linear program to solve for the ORC right? Please note that the discussion paragraph added on page 10 lines 277-288 should specify what m and n are representing, i.e., number of edges, number of nodes, etc. Also, it would be good to provide intuition to why the particular $O(mn^{5/2})$, since the discussion misses the convention or definitions of the notations, I cannot help to provide a correct O complexity result. I appreciate the authors' efforts but this needs a bit more precision in notation.- It is unclear from the response and the manuscript what the authors mean by multiscale. I do agree with the authors that neither of the prior Ollivier-Ricci community detection algorithms (i.e., Sia et al., Sci Rep 2019 and Ni et al Sci Rep 2019) do not address multiscale, but this should be mentioned for both I believe. Of note, some of the references the authors mention like 57 already exploited multiscale and self-similarity for community detection as well.
- I would have appreciated if in each answer precise explanations on how text / discussion has been changed and what has been added. I consider all of the above minor issues and so I recommend this manuscript for acceptance.

Reviewer #3:

Remarks to the Author:

Thank you for your revisions. This is a well-written paper, and I have no further requests. It should be accepted for publication.

We have addressed the comments of Reviewer 2 as described below. We have also addressed the editorial requests and highlighted the changes in the manuscript and the supplementary material in the attached diff.pdf and diff_si.pdf files, respectively.

We hope that our manuscript is now formally acceptable for publication.

Kind Regards,

Adam Gosztolai and Alexis Arnaudon

Reviewer #2

I read the revised manuscript and the response letter and the changes made by the authors are substantial and commendable. I summarize below a few minor issues:

- In this response "In addition, we extended our literature review and more explicitly clarified the connection to related classes of methods.", I would have liked to see how the discussion in the manuscript has been enriched.

We have now extended the Discussion in Lines 373-378 to reflect on the performance comparison with other methods.

Similar comment applies for this reviewer's comment "I would encourage the authors to do a full literature search and compare with prior methods as well." and response from the authors "Thank you for this feedback. We have now extended our literature review in the Introduction." where it would have been useful to learn what is the newly added information and how it changed the manuscript.

We have addressed all these points in the previous revision. We noticed that we referenced the wrong Supplementary Note in lines 274-275, which may have caused the confusion. In this revised version, we have corrected the reference to the Supplementary Figures and Notes. We have also sharpened the text in Supplementary Note 4.

To summarise, we have

- 1) Briefly surveyed the multiscale community detection literature in lines 53-58 and 62-64 in the Introduction.
- 2) Compared our 'geometric clustering' method with state-of-the-art algorithms from major classes of clustering methods in generative benchmark graphs (see lines 272-277, Supplementary Figure 2 and details in Supplementary Note 3).

3) Compared against multiscale clustering methods on the *C. elegans* dataset (see lines 323-327, Supplementary Figure 3 and details in Supplementary Note 4).

Similarly for this comment of reviewer 2 "13) The authors mention in the abstract and discussion that the curvature and embedding dimension can be used to ..." for which the authors responded " We agree that this is a research direction that our work will likely have the largest impact on. Motivated by this comment we elaborated in the Discussion on possible research directions concerning the study of synchronisation problems and chimera states by identifying information limiting edges." Of note, the control or tuning of the network either by adding or removing links or by adjusting the edge weights goes beyond synchronization and can deal with controllability of complex networks. I believe that their proposed work can be well aligned with recent efforts on controlling multifractality of networks but this needs to be carefully discussed. In general, changes in revisions should have been highlighted.

We have briefly mentioned the possibility to use our methods for controlling the multifractal geometry on Line 358-359. However, as we have not explored this direction we kept this discussion to a minimum.

Please note that reference [28] is cited, but it is unclear where the comparison exists.

The comparison with the method of Tremblay and Borgnat (Ref. [28]) on the *C. elegans* dataset is discussed in Supplementary Note 4. Beyond this numerical comparison we have not analysed the theoretical connection between the methods because apart from the use of diffusions the construction of the two methods is quite different.

- I appreciate the comparison with the graph wavelet and the other method suggested by the reviewer, it would be nice if the codes are made available. Please explain where readers can find the code.

We uploaded all code and data as detailed under the 'Code availability' and 'Data availability' statements.

- Regarding comment 4 of reviewer 2 on computational complexity, the response doesn't shed light on the computational complexity, I assume it is bounded by the computational complexity of the linear program to solve for the ORC right?

Given a set of diffusion measures, the theoretical complexity of computing the curvatures is at most $O(mn^{5/2})$. The Reviewer is correct that, intuitively, one needs to solve m -times a linear program (transportation problem) of complexity $O(n^{5/2})$. We also note that the computation also involves obtaining the shortest path distances and the diffusion measures. Although the complexity of the latter is not known, as we show in Supplementary Note 5 and Supplementary Figure 4 the computation of the curvatures is the limiting factor in practice.

We have clarified this intuition in the main text in lines 383-384.

Please note that the discussion paragraph added on page 10 lines 277-288 should specify what m and n are representing, i.e., number of edges, number of nodes, etc.

We have defined n and m in Line 80 during the initial problem setup and, indeed, they refer to the number of nodes and edges, respectively.

Also, it would be good to provide intuition to why the particular $O(mn^{5/2})$, since the discussion misses the convention or definitions of the notations, I cannot help to provide a correct O complexity result. I appreciate the authors' efforts but this needs a bit more precision in notation.

We have explained the intuition in the response above. We have now clarified the $O()$ notation on lines 282-283.

It is unclear from the response and the manuscript what the authors mean by multiscale. I do agree with the authors that neither of the prior Ollivier-Ricci community detection algorithms (i.e., Sia et al., Sci Rep 2019 and Ni et al Sci Rep 2019) do not address multiscale, but this should be mentioned for both I believe. Of note, some of the references the authors mention like 57 already exploited multiscale and self-similarity for community detection as well.

In our work multiscale structure refers to the existence of clusters at different distinguishable resolutions. We clarified this on line 33.

I would have appreciated if in each answer precise explanations on how text / discussion has been changed and what has been added. I consider all of the above minor issues and so I recommend this manuscript for acceptance.